# Telomere damage induces internal loops that generate telomeric circles

Giulia Mazzucco[1], Armela Huda[1], Martina Galli[1], Daniele Piccini[1], Michele Giannattasio [1,2], Fabio Pessina[1] & Ylli Doksani [1✉]

Extrachromosomal telomeric circles are commonly invoked as important players in telomere maintenance, but their origin has remained elusive. Using electron microscopy analysis on purified telomeres we show that, apart from known structures, telomeric repeats accumulate internal loops (i-loops) that occur in the proximity of nicks and single-stranded DNA gaps. I-loops are induced by single-stranded damage at normal telomeres and represent the majority of telomeric structures detected in ALT (Alternative Lengthening of Telomeres) tumor cells. Our data indicate that i-loops form as a consequence of the exposure of single-stranded DNA at telomeric repeats. Finally, we show that these damage-induced i-loops can be excised to generate extrachromosomal telomeric circles resulting in loss of telomeric repeats. Our results identify damage-induced i-loops as a new intermediate in telomere metabolism and reveal a simple mechanism that links telomere damage to the accumulation of extra-chromosomal telomeric circles and to telomere erosion.

[1] IFOM, the FIRC Institute of Molecular Oncology, via Adamello 16, 20139 Milan, Italy. [2] Dipartimento di Oncologia ed Emato-Oncologia, Università degli Studi di Milano, Via Festa del Perdono 7, Milan 20122, Italy. ✉email: ylli.doksani@ifom.eu

Mammalian telomeres are made of several kilobases of tandem TTAGGG repeats that are required to protect chromosome ends from the DNA damage response. Erosion of telomeric repeats can lead to senescence and genome instability and, therefore, plays important roles in ageing and tumorigenesis[1]. Extrachromosomal circular DNAs made of telomeric repeats (t-circles) have been found in a wide range of organisms and are thought to play dual opposing roles in telomere maintenance. On one hand they have been associated to telomere loss via deletion/trimming of telomeric repeats, while in different contexts, such as ALT cells they could promote telomere elongation through rolling-circle amplification[2–5]. C-circles (t-circles with a covalently closed and partially single-stranded C-rich strand) accumulate in ALT cells and provide a diagnostic marker for ALT tumors[6]. Despite their relevance in telomere biology, it is not clear how these circles are generated. Telomeric circles have been detected in cells expressing a TRF2 mutant that lacks the N-terminal basic domain and, given the role of TRF2 in the formation/maintenance of t-loops, it has been proposed that t-circles could form via nucleolytic excision of the t-loop structure[2,7–9]. However, more recently, t-circles have also been found in normal cells and in an ever-growing list of mutants, apparently unrelated to t-loop metabolism[3,10–17]. These results suggest the existence of additional mechanisms of t-circle formation. T-circles can be detected through a rolling-circle replication assay and their presence is often inferred from the appearance in two-dimensional agarose gel electrophoresis (2D-gels) of an arc compatible with the migration of relaxed circular DNA[2,10,18]. Telomeric circles have been found in electron microscopy (EM) images of ALT telomeres[5], but further analysis (e.g., identification of intermediates of t-circle formation) has been hampered by the inconsistency of available procedures for telomere purification.

Using a newly-developed telomere purification procedure, combined with EM analysis, we found that damaged telomeres tend to form internal loops (i-loops), likely due to the exposure of single-stranded DNA at telomeric repeats. These structures migrate in the t-circle arc of 2D-gels and represent the majority of telomeric structures found in ALT cells. We show that damage-induced i-loops can be excised as telomeric circles, resulting in telomere loss. These results identify damage-induced i-loops as a key intermediate in telomere circle formation and provide a mechanism that links telomere damage with t-circle formation and telomere erosion.

## Results

### A two-step procedure for the purification of mammalian telomeres.
Telomeric repeats lack restriction sites and this property has been exploited for their enrichment by digestion of non-telomeric DNA with frequent cutters and then purification of large DNA fragments, containing telomeres, through gel filtration columns[19,20]. Key telomere features (e.g., t-loops, t-circles) have been visualized with this approach; however, incomplete digestion of non-telomeric DNA and poor fractionation of milligrams of DNA by the gel filtration columns have limited its applications. To overcome these issues, we developed a two-step procedure for the large-scale purification of telomeric repeats from mammalian cells. First, 2.5 mg of mouse genomic DNA are digested with frequent cutters and separated in a sucrose gradient (Fig. 1a). Then, high molecular weight fractions containing the telomeric repeats are collected, digested again with a new mixture of restriction enzymes (see materials and methods), and separated in a preparative agarose gel (Fig. 1b). The high molecular weight DNA recovered from the agarose gel shows a ~1000-fold increase in telomeric repeats compared to the starting material, while

more abundant mouse long interspersed repeats (L1 repeats) are undetectable (Fig. 1c). Telomere enrichment was confirmed in single-molecule IF-FISH analysis, where over 80% of the DNA molecules from enriched samples are recognized by a telomeric probe, while <1 in 1000 telomeric fibers were present in non-enriched samples (Fig. 1d). We obtained similar enrichment levels from human cells with long telomeres (Supplementary Fig. 1a, b).

### Frequent i-loops in telomere-enriched samples.
We employed the procedure described above to isolate telomeres from mouse embryo fibroblasts (MEFs) and analyze their structure by EM. The DNA was crosslinked with psoralen in vivo, prior to cell lysis, and the telomere-enriched material was spread with the BAC method and rotary shadowed with platinum[21]. In telomeric spreads, DNA fragments ranged from 2 to 40 kb. As expected, telomeric samples were enriched in t-loops, although their absolute frequency in our spreads was lower than in previous settings (Supplementary Fig. 2a–c)[20,22].

One salient feature we observed in telomeric spreads was the occurrence of molecules with i-loops (Fig. 2a). Differently from t-loops, which sequester one end of the DNA molecule and are therefore terminal, i-loops appeared as crossings of the internal regions of the molecules, where the ends are not engaged. In three independent experiments, with SV40 Large T antigen (LT)-immortalized MEFs, around 14% of molecules in the telomere-enriched samples had one or more i-loops. In control spreads of genomic DNA, fragmented at a similar size by restriction digestion, i-loops occurred in around 3% of the molecules (Fig. 2b). Accumulation of i-loops at telomeres could not be attributed to the enrichment procedure as mouse genomic DNA subjected to a mock enrichment procedure (where the restriction enzymes were omitted) showed i-loops in 5.1% of molecules ($N = 927$ molecules). An abundance of molecules with i-loops was seen also in telomeric spreads from HeLa 1.3 cells with long telomeres (9.2% vs 2.9%; Supplementary Fig. 3a–c). Accumulation of i-loops at telomere-enriched samples was also observed when the DNA was spread with the Kleinschmidt method[23], although the background level of internal loops was higher in this setting (92.9% vs 60.3%; Supplementary Fig. 4a–d).

I-loops ranged from 0.2 to 25 kb, with a median size of 1.6 kb (Fig. 2c). In the majority of cases i-loops occurred once per molecule, but in about 25% of cases, two or more i-loops were present on the same molecule (Fig. 2d). When we examined the structure of the i-loops at higher magnification, we noticed that about one in four had a thinner, apparently single-stranded, region at the junction (Fig. 3). In another 20% of the loops, a short gap and/or a small flap was visible in one of the DNA strands near the junction. In bulk genomic DNA samples, around 18% of the loops showed a single-stranded region at (or near) the loop junction ($N = 109$ loops). Based on these observations, we hypothesized that i-loops could represent structural transitions that occur at sites of single-strand damage (i.e., nicks and gaps) on the telomeric repeats.

### I-loops are the majority of structures detected at ALT telomeres.
Since i-loops often occurred in proximity of single strand damage, we turned our attention to ALT cells, which contain nicks and gaps at telomeres and show unusual telomeric structures in 2D-gels[2,24]. In particular, a faint, slow-migrating arc is detected in 2D-gels at ALT telomeres; this signal is commonly known as the t-circle arc and is attributed to the presence of extrachromosomal telomeric circles[2,5,18,24]. Although the t-circle arc is compatible with the migration of relaxed circular DNA, there is no direct evidence on the types of telomeric structures

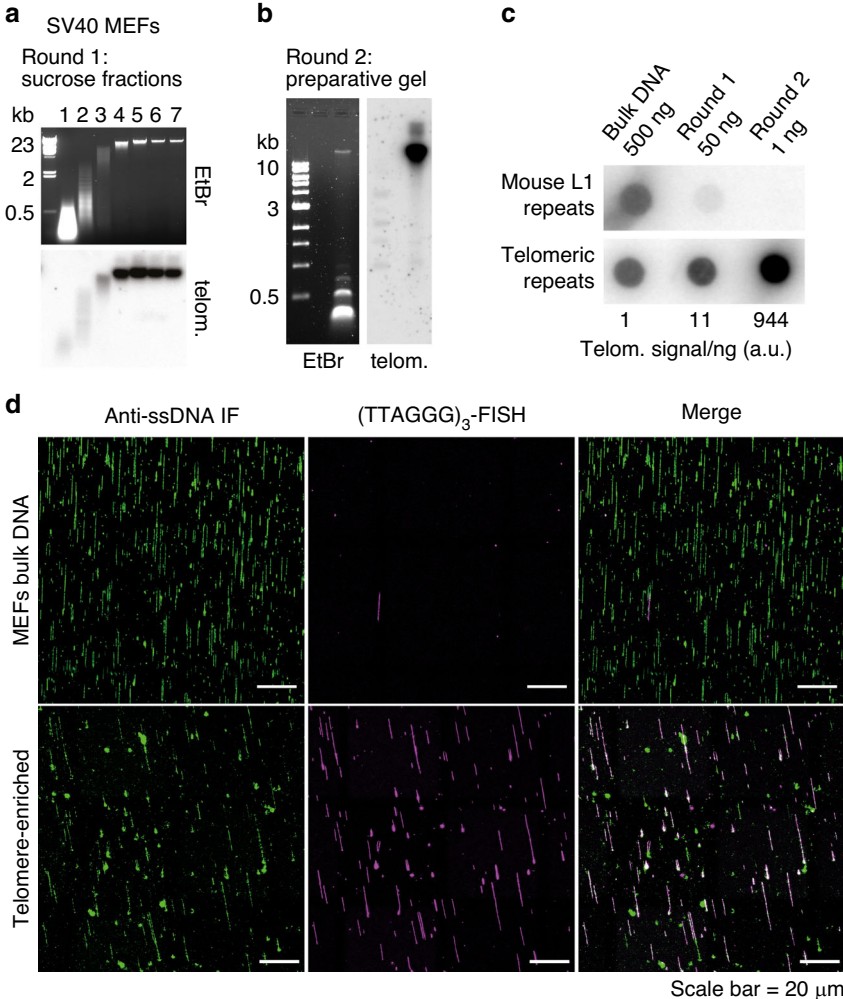

**Fig. 1 A two-step procedure for the purification of mammalian telomeres. a** Top: agarose gel showing the separation of the large telomeric repeat fragments from the bulk genomic DNA in a sucrose gradient. Genomic DNA (∼2.5 mg) from SV40LT-immortalized MEFs was digested with HinfI and MspI. The digested DNA was separated by centrifugation on a sucrose gradient. Seven fractions were collected and an aliquot (∼1/500) of each fraction was loaded on an agarose gel. Bottom: the gel was blotted onto a membrane and hybridized with a TTAGGG repeats probe to verify that telomeric repeats remained in the high molecular weight (HMW) fractions. Source data are provided as a Source Data File. **b** Left: agarose gel showing the separation of the large telomeric repeat fragments from the remaining non-telomeric DNA, in the second purification round. The HMW DNA, contained in the last four fractions of the sucrose gradient described in (a), was recovered and digested with RsaI, AluI, MboI, HinfI, MspI, HphI, and MnlI. The digested DNA was separated on a preparative agarose gel and the DNA migrating in the area above 5 kb was extracted from the gel. The image shows an aliquot (∼1/100) of the digested DNA, separated on an agarose gel. Right: the gel was blotted onto a membrane and hybridized with a TTAGGG repeats probe to verify that telomeric repeats remained in the HMW area. Source data are provided as a Source Data File. **c** Dot blot analysis showing the enrichment of telomeric repeats. The indicated amounts of DNA from each enrichment step were spotted on a membrane and hybridized either with a probe recognizing the long interspersed L1 repeats or TTAGGG repeats. The amount of TTAGGG repeat signal/ng was quantified and reported relative to the signal/ng value in the initial, non-enriched DNA. Source data are provided as a Source Data File. **d** Single-molecule analysis showing the enrichment of the telomeric repeats. The DNA was combed onto silanized coverslips, denatured in situ and labeled sequentially with an antibody against single-stranded DNA and a Cy3-labeled (TTAGGG)$_3$ PNA probe.

that populate it. We decided to purify the DNA molecules from the t-circle area of the 2D gel in order to visualize their structure by EM. Genomic DNA was prepared from U2OS cells and telomeres were enriched with the procedure described above, except that in the second round of enrichment the DNA was separated in a 2D-gel (Fig. 4a, Supplementary Fig. 5a). The areas of the second-dimension gel containing the t-circle arc and the linear telomeres were excised (Fig. 4a), the DNA was recovered and analyzed by EM. As expected, the material purified from the t-circle area was richer in DNA structures (including i-loops, t-loops, circles and, at lower frequencies, Y-shaped and X-shaped molecules), although it still contained substantial amounts of linear fragments, likely deriving from resolved structures and/or

imperfect separation in the 2D-gels (Supplementary Fig. 5b). Around 9% of the molecules were circular and, in 28% of these circles, single-stranded gaps were visible (Fig. 4b, c). This result confirms previous reports on the presence of double-stranded and partially single-stranded telomeric circles in ALT cells[2,5,6]. However, molecules with one or more i-loops represented 40% of all DNA recovered from the t-circle area, over 4-fold more abundant than telomeric circles (Fig. 4b, d, Supplementary Fig. 5b). Therefore, i-loops represent the vast majority of telomeric structures identified by 2D-gels in ALT cells. Moreover, similar to mouse telomeres, close inspection of i-loops in ALT telomeres revealed that they often occurred in proximity of strand damage (red arrows in Fig. 4d).

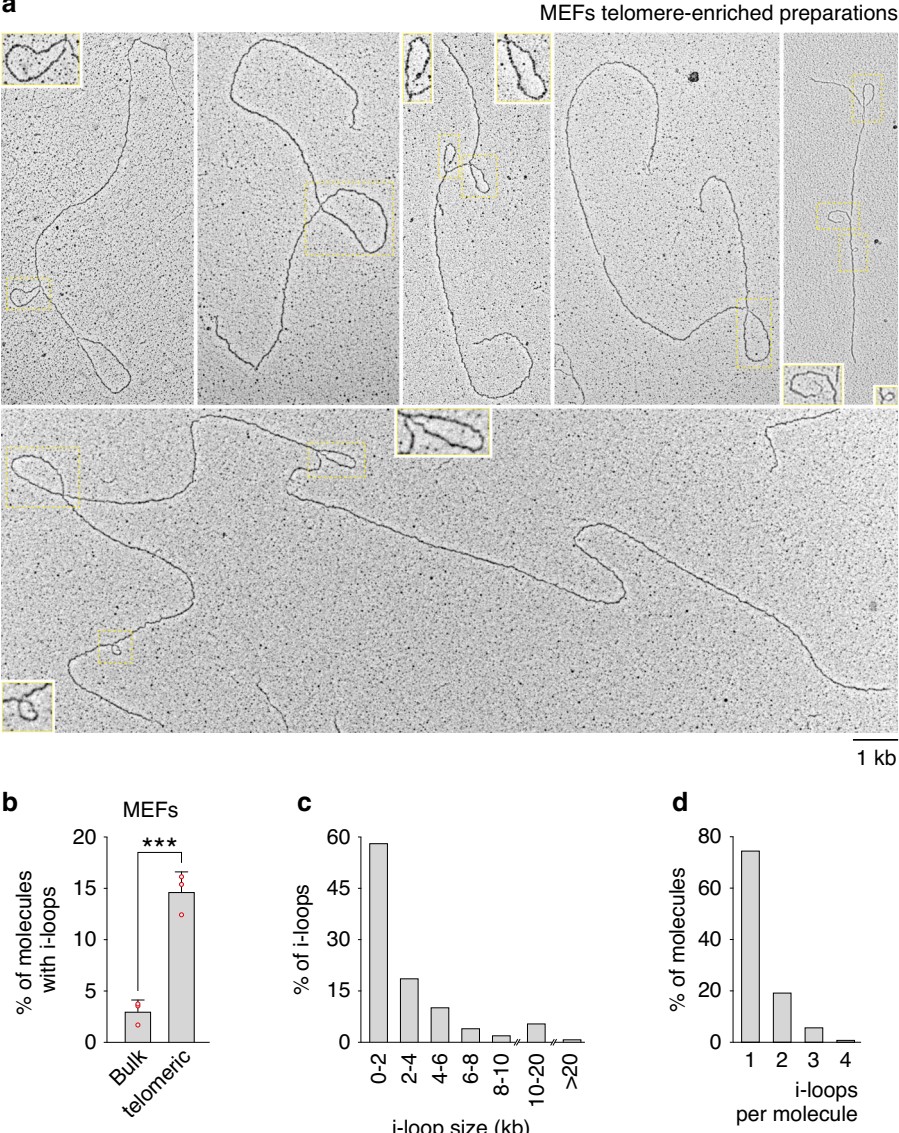

**Fig. 2 Frequent i-loops in telomere-enriched samples. a** Examples of molecules with i-loops found in the telomeric samples. Telomeric DNA, enriched with the procedure described in Fig. 1, was analyzed in transmission Electron Microscopy (EM). I-loops are indicated by the yellow rectangles. Insets show 2X enlargements of some of the areas inside the yellow rectangles. **b** Quantification of i-loop occurrence. Telomere-enriched (telomeric) and non-enriched genomic DNA (bulk) were analyzed in EM as described above. The non-enriched, bulk DNA was digested with KpnI, which generates fragments of 10 kb in average. The fraction of molecules containing one or more i-loops was scored in 3 independent experiments ($n = 310$, 715, and 845 molecules for telomere-enriched samples and 1466, 776, and 1776 molecules for non-enriched samples). Bars represent the mean with the standard deviation. Single data points are also shown as red dots. *P* value = 0.0009, was derived from unpaired, two-tailed, Student's *t*-test. Source data are provided as a Source Data File. **c** Size distribution of the i-loops in the telomeric DNA. *N* = 342 i-loops. Source data are provided as a Source Data File. **d** Distribution of the number of i-loops per molecule in the telomeric DNA. *N* = 266 molecules with i-loops. Source data are provided as a Source Data File.

**I-loops are induced by single-strand damage at telomeric repeats**. Since i-loops were associated with single-stranded telomere damage and populated the t-circle arc in U2OS cells, we asked whether telomere damage alone can induce their formation and therefore the appearance of the t-circle arc in 2D-gels. To test this hypothesis, we used mild DNase I treatment that introduces both nicks and short single-stranded gaps on DNA[25]. Following the incubation of MEFs nuclei with increasing concentrations of DNase I, genomic DNA was isolated, digested with frequent cutters and separated in 2D-gels. Telomeres from mock-treated nuclei migrated mainly as linears with no t-circle arc visible, while damaged telomeres, isolated from nuclei that were treated with DNase I, showed a strong accumulation of the t-circle arc

(Fig. 5a). We obtained the same result in human cell lines with long or short telomeres, although t-circle arc induction strongly decreased with telomere length (Supplementary Fig. 6a, b). No arc was induced by the abundant mouse L1 repeats, or by the bulk genomic DNA, showing that, at these magnitudes, this is not a general feature of nicked DNA (Supplementary Fig. 6c).

Since the DNase I treatment was performed on isolated nuclei, we asked whether the chromatin environment or any chromatin-associated factor is required for the generation of the structures migrating in the t-circle arc. Surprisingly, mild DNase I treatment of isolated, protein-free, genomic DNA resulted in a strong induction of the t-circle arc at telomeric repeats, while the same structural transition was not observed in the bulk genomic DNA,

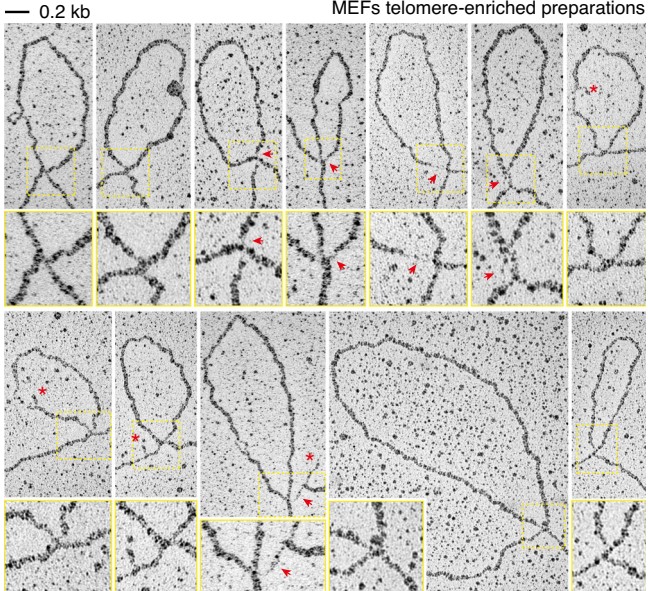

— 0.2 kb　　　　　　　　　MEFs telomere-enriched preparations

**Fig. 3 I-loops often occur in proximity of strand damage.** High magnification images of i-loops observed in telomere-enriched fractions of the experiment described in Fig. 2. A 2X enlargement of the area inside the yellow rectangle is shown under each loop. Red arrows indicate regions of single-stranded DNA at the loop junction, while red asterisks indicate small flaps in proximity of the loop junction.

or at the L1 repeats (Fig. 5b, Supplementary Fig. 6d). The absence of specialized enzymatic activities in this setting rules out the accumulation of telomeric circles (see also below), demonstrating that the 2D-gel arc, commonly identified as the t-circle signal, can be generated in the absence of telomeric circles.

Having established that telomere strand damage is sufficient for the induction of the t-circle arc, we verified that this process was indeed associated with the accumulation of telomeric i-loops, as predicted. The same DNase I treatment, as described above, was performed in large scale; then, genomic DNA was isolated, processed through the two-step procedure for the enrichment of telomeric repeats and analyzed by EM (Fig. 5c). Telomere strand damage induced by DNase I resulted in a threefold increase in i-loops, while in the bulk genomic DNA control, i-loops remained at basal levels (Fig. 5d, e). Together, these experiments show that accumulation of nicks and gaps at telomeric repeats is sufficient to induce i-loops and generate the t-circle signal in 2D-gels.

Based on these experiments, we hypothesized that i-loops are formed via spontaneous annealing and branch migration events, occurring at sites of gaps or nicks favored by the extremely high abundance of homology at telomeric repeats. If this is the case, then formation of i-loops after the induction of DNA damage should be limited by inter-strand psoralen crosslinking that prevents DNA branch migration and preserves native DNA structures[26]. To test this hypothesis, we first performed psoralen crosslinking on the nuclei, before the DNase I treatment and then analyzed telomere structures in 2D-gels, as above. DNase I damaged to a similar extent both crosslinked and non-crosslinked DNA; however, formation of i-loops was strongly inhibited by psoralen crosslinking, as seen by the reduced intensity of the t-circle arc in 2D-gels (Fig. 6a). This result suggests that i-loop formation, after the induction of telomere strand damage, requires DNA branch migration. Importantly, once formed, i-loops are not sensitive to crosslinking. Indeed, the t-circle arc of U2OS cells was not affected by psoralen crosslinking (Fig. 6b), showing that ALT telomeres, which experience endogenous damage, contain i-loops in vivo, prior to psoralen crosslinking.

**I-loops are a substrate for the generation of telomeric circles.** The short TTAGGG telomere repeat motif provides a context where almost any single-stranded gap exposes intramolecular homology. For instance, two gaps on opposite telomeric strands could generate an i-loop simply by strand annealing (Fig. 7a). Further branch migration and strand exchange events could generate a double Holliday junction (HJ) at the base of the i-loop. Similarly, two gaps on the same telomeric strand could undergo strand exchange and generate i-loops with a single HJ at the base (Fig. 7b). We hypothesized that these damage-induced i-loops would be a substrate for HJ resolvases, in a reaction similar to the one proposed for unprotected t-loops[2,7]. In this resolution reaction 50% of events would result in telomere deletions and generation of extrachromosomal telomeric circles (Fig. 7c). In order to test whether i-loops are biochemically a substrate for the generation of telomeric circles, we first treated mouse genomic DNA with DNase I to induce i-loops at telomeres, and then incubated the DNA with a nuclear extract from HeLa cells (Fig. 7d). The DNA was then recovered and subjected to the rolling-circle replication assay, in the absence of a telomeric primer[6]. Neither DNase I treatment nor incubation with the nuclear extract alone induced telomeric circles, while the combination of the two resulted in a fivefold increase in circles (Fig. 7d–f). Importantly, accumulation of telomeric circles required both Mg++ and ATP, consistent with the requirements of a Holliday junction resolution reaction. This experiment shows that i-loops are a substrate for the generation of extrachromosomal telomeric circles, in a reaction resembling HJ resolution.

## Discussion

We identify damage-induced i-loops as key intermediates that link telomere damage to telomere erosion and the generation of extrachromosomal telomeric circles. Our data provide an additional mechanism of t-circle formation, as a consequence of telomeric damage. These results predict that conditions associated with chronic telomere (or DNA) damage (e.g., chemotherapeutics, replication stress) will favor the formation of telomeric circles and promote telomere loss, while factors that prevent formation of i-loops at sites of damage (e.g., factors that prevent strand exchange or improper single-strand annealing at telomeres) would counteract the accumulation of extrachromosomal telomeric circles. Given that ALT cells are known to experience endogenous telomere damage, the mechanism proposed in Fig. 7 could help explain the continuous generation of telomeric circles in ALT cells. In the same view, accumulation of telomeric damage, could be a common denominator that explains the presence of t-circles in many mutants of genes involved in DNA metabolism and telomere maintenance[11–15]. Frequent formation of i-loops could provide yet another challenge to replication fork progression at telomeric repeats and contribute to telomere fragility[27,28]. I-loops could be a relevant substrate for specialized helicases, such as Rtel1, Blm, and Wrn, which could prevent formation or promote branch migration/dissolution of i-loops at telomeric repeats, thereby reducing the probability of telomere loss due to i-loop excision[29–31]. This process could be hindered in ALT cells due to the presence of non-canonical telomeric repeats[32].

Damage-induced i-loops might also occur at other tandem repeats, explaining the formation of circular DNAs at these sequences from yeast to human[18,33,34]. In this view, it is important to notice that the overall rate of i-loop formation will be higher at repetitive elements with a shorter repeated motif, because they will be more likely to expose complementary sequences when damaged. Therefore, telomeres, with a repeat

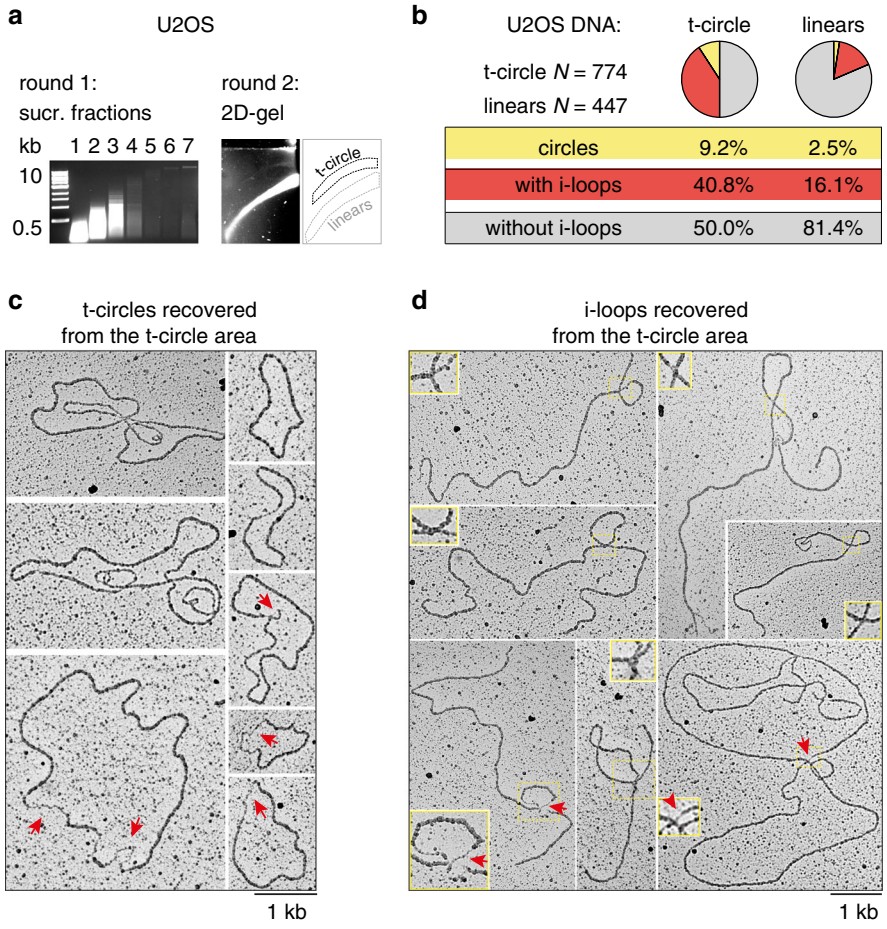

**Fig. 4 I-loops are the majority of telomeric structures detected at ALT telomeres. a** Procedure for the purification of telomeric DNA migrating in the linear and the t-circle arc of ALT telomeres. Genomic DNA (~2.5 mg) extracted from U2OS cells was processed for the telomere enrichment procedure as described in Fig. 1a. The HMW DNA, contained in the last 4 fractions of the sucrose gradient, was collected, digested again as described in Fig. 1b and separated in a 2D-gel. A strong linear signal and a faint t-circle arc were visible in the second-dimension gel (right). These areas were excised and the DNA was recovered from the gel. Source data are provided as a Source Data File. **b** Pie chart showing the distribution of the molecules recovered from the 2D-gel. Percentages of the 3 major categories are shown. Note that i-loops can occur also in molecules having a t-loop at the end, or at branched molecules. This sub-distribution is reported in Supplementary Fig. 5b. **c** Example of circular molecules found in the DNA purified from the t-circle arc. Arrows indicate regions of single-stranded DNA. **d** Examples of i-loops found in the DNA purified from the t-circle arc. Insets represent 2X enlargements of the areas inside the yellow rectangles. Red arrows indicate regions of single-stranded DNA at the loop junction.

unit of 6 nt, will be more prone to generate extrachromosomal circles compared to most other long repeats. This high propensity of telomeric repeats to form i-loops that can be excised as circles would result in continuous and stochastic variations in the number of repeats thus explaining, at least in part, the amplitude of telomere length heterogeneity across different chromosomes or different cells.

A positive correlation between telomere length and accumulation of the t-circle signal in 2D-gels has been reported in normal and stem cells, indicating the existence of a trimming mechanism that controls telomere length[3,35]. Our results suggest that, as telomere length increases so will the probability of i-loop formation and excision due to stochastic damage. This correlation could be relevant in understanding the sources of dysfunctional telomeres and how telomere length evolves in different organisms.

## Methods

**Cell culture**. SV40LT-immortalized MEFs were grown in D-MEM (Lonza, BE12-614F) supplemented with 10% fetal bovine serum (EuroClone, ECS0180L), 2 mM L-glutamine (EuroClone, LOBE17605F), 100 U/ml penicillin-0.1 μg/ml

streptomycin (EuroClone, ECB3001L), 0.1 mM non-essential amino acids (Microtech, X-0557). U2OS cells (ATCC) were grown in Mc Coy's 5 A w/Glutamax (Life Technologies, 36600-088) supplemented with 10% fetal bovine serum (EuroClone, ECS0180L). U2OS cells were authenticated using the GenePrint® 10 System (10-Locus STR System for Cell Line Authentication) by Promega CAT. NUM. B9510. HeLa 1.3, HeLa204 and HTC75 cells were grown in D-MEM (Lonza, BE12-614F) supplemented with 10% fetal bovine serum (EuroClone, ECS0180L), 2 mM L-glutamine (EuroClone, LOBE17605F), 100 U/ml penicillin-0.1 μg/ml streptomycin (EuroClone, ECB3001L). HeLa1,3, HeLa204, and HTC75 cell lines are a gift from Titia de Lange. All cell lines are tested for mycoplasma both upon arrival at IFOM and after a new stock of cells is made, and all of them resulted to be negative for mycoplasma contamination. Mycoplasma test is performed by the IFOM Cell Biology UNIT and consist of two independent tests: a PCR analysis (For detail protocol see[36] and a biochemical test (MycoAlert Detection Kit, Lonza Catalog #: LT07-418).

**Enrichment of telomeric repeats**. Around $500 \times 10^6$ cells, were harvested and resuspended in ice-cold PBS. For psoralen crosslinking, the cell suspension was poured in a 10 cm dish and kept on ice while stirring, throughout the procedure. The suspension was first incubated with 30 μg/ml 4, 5′, 8-trimethylpsoralen (Sigma, T6137, stock 2 mg/ml in DMSO, stored at −20 °C) for 5 min in the dark and then exposed to 365 nm UV light for 8 min in a UV Stratalinker 1800, (Stratagene), with 365 nm UV bulbs (model UVL-56, UVP) at 2–3 cm from the light source. The incubation and irradiation steps were repeated three more times (4 cycles total). Cells were then lysed in TNES buffer (Tris 10 mM pH8.0, NaCl 100 mM, EDTA

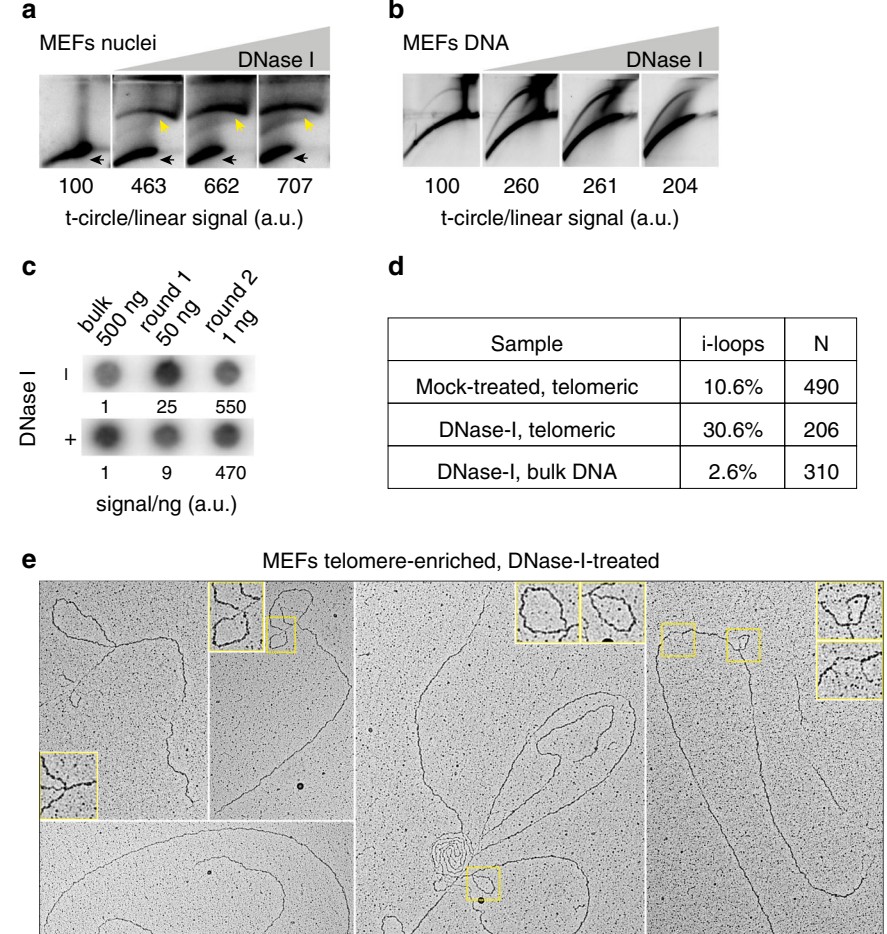

**Fig. 5 I-loops are induced by strand damage at telomeric repeats (see also Supplementary Fig. 6). a** 2D-gels showing that the t-circle arc can be strongly induced by formation of nicks and gaps. MEFs nuclei were incubated with 0; 1; 2.5 or 5 μg/ml of DNase I for 8 min at RT. The reaction was stopped, the DNA was separated in 2D-gels, blotted on a membrane and hybridized with a telomeric probe. The signal ratio in the t-circle arc (yellow arrows) and in the linears (black arrows), is reported relative to the untreated sample, which was arbitrarily set to 100. Source data are provided as a Source Data File. **b** 2D-gels showing that the t-circle arc can form spontaneously, in the presence of nicks and gaps at telomeres. Isolated mouse DNA was incubated with 0; 0.1; 0.2 or 0.4 μg/ml of DNase I for 8 min at RT. The reaction was stopped, the DNA was separated in 2D-gels, blotted on a membrane and hybridized with a telomeric probe. The ratio of the signal in the t-circle arc and in the linears is reported relative to the untreated sample, which was arbitrarily set to 100. Source data are provided as a Source Data File. **c** Dot blot showing the enrichment of telomeric repeats, after large-scale DNase I treatment. Around 500 × 10⁶ SV40LT-immortalized MEFs nuclei were incubated either with 0 or 5 μg/ml of DNase I for 8 min at RT. The reaction was stopped and telomeres were enriched with the procedure described in Fig. 1. The indicated amounts from each enrichment step were spotted on a membrane and hybridized with a telomeric probe. The signal per ng of DNA is reported relative to the non-enriched DNA. Source data are provided as a Source Data File. **d** Accumulation of i-loops at telomeres damaged by DNase I. Telomere-enriched DNA from the experiment described in (**c**) was analyzed in EM. The percentage of molecules containing i-loops is reported. A KpnI-digested bulk genomic DNA control was included for the sample treated with DNase I. **e** Examples of molecules with i-loops observed at telomere preparations from DNase I-treated nuclei. Insets show 2X enlargements of the area inside the yellow rectangles.

10 mM; 0.5% SDS) incubated with 50 μg/ml RNaseA (Sigma, R500) for 60 min at 37 °C, and then with 100 μg/ml Proteinase K (Roche, 3115887001) for 12 h at 37 °C. The DNA was extracted with Phenol Chloroform Isoamyl alcohol 25:24:1 (Sigma, P2069) followed by an extraction with Choloroform (VWR, 22711) and precipitation with isopropanol. Around 2.5 mg of DNA was digested overnight with 750 units of HinfI and MspI (NEB). The digestion was precipitated and loaded on a sucrose gradient, 10%–20%–30% sucrose, 8 ml each fraction, in TNE buffer and centrifuged in SW32-Ti rotor (Beckman) at 30100 rpm (111265 g) for 16 h. The HMW fractions containing the telomeric repeats were collected, concentrated, and washed twice with Tris 10 mM pH 8.0 in Amicon Ultra-15 Ultracel-PL PLTK, 30 kDa MWCO (Millipore/MERCK UFC903024) filters. The DNA was then digested overnight with 50 units each of RsaI, AluI, MboI, HinfI, MspI, HphI, MnlI (NEB), and then separated on a 0.7% low-melting agarose gel (SeaPlaque Agarose, Lonza, 50100), without ethidium bromide. Fragments migrating above the 5 kb

band of the marker were extracted using the Silica Bead DNA gel extraction kit (Thermo Fisher Scientific, K0513) following the manufacturer's instructions, except that once the DNA was bound, the beads were not resuspended to avoid mechanical shearing of the DNA. The DNA was eluted in TE 1X and quantified using Qubit dsDNA HS assay kit (Invitrogen, Q32854). Ethidium Bromide gel acquisitions were performed with a Chemidoc XRS + Imaging system and Image Lab software (v3.0, Bio-Rad Laboratories).

**Single-molecule analysis of telomeric enrichment**. Around 10 ng of bulk genomic DNA or telomere-enriched DNA was combed on silanized coverslips (Genomic Vision, COV-002) using the DNA Fiber Comb apparatus (Genomic Vision, version 3 REF: MSC-001). The coverslips with the DNA were subjected to the following treatments: baking for 2–3 h at 60 °C, denaturing in 0.5 M NaOH, 1

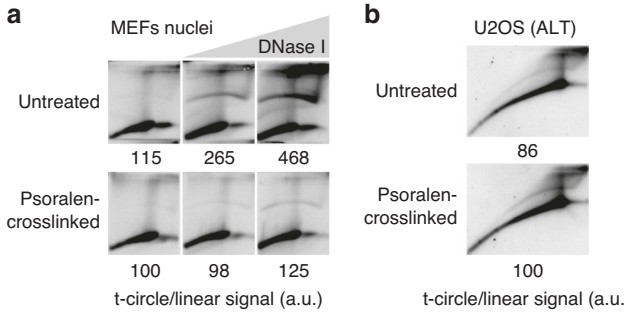

**Fig. 6 I-loop formation after DNA damage requires branch migration. a** A preparation of SV40LT-immortalized MEFs nuclei was split in two and one half was psoralen crosslinked on ice (in this prep, DNA branch migration is largely prevented). Then, both preps were treated with DNase I and processed for 2D-gels as described in Fig. 5a. The ratio of the telomeric signal in the t-circle arc and in the linears is reported relative to the untreated and crosslinked sample, which was arbitrarily set to 100. Source data are provided as a Source Data File. **b** 2D-gel analysis showing that the i-loops that accumulate in ALT cells are not affected by psoralen crosslinking. U2OS cells were psoralen crosslinked on ice, then genomic DNA was extracted and processed for 2D-gels as above. The ratio of the telomeric signal in the t-circle arc and in the linears, is reported relative to the crosslinked sample which was arbitrarily set to 100. Source data are provided as a Source Data File.

M NaCl for 8 min, followed by two washes in PBS and dehydration in ethanol series (70%; 90%; 100%, 1 min each). Blocking with 5% BSA in PBS for 1 h at 37 °C and incubation with an anti-single-stranded DNA antibody (Sigma, MAB3034) diluted 1:80 in 5% BSA in PBS, for 2 h at RT, followed by three washes with PBS-0.05% Tween 20. Incubation with Alexa488-labeled anti-mouse secondary antibody (Invitrogen, A1101) diluted 1:400 in 5% BSA in PBS, followed by three washes with PBS-0.05% Tween 20 and dehydration in ethanol series (70%; 90%; 100%, 1 min each). Incubation with a Cy3-labeled TTAGGG$_3$ PNA probe (PNA Bio, F1006), 50 nM in 70% formamide (Thermo Scientific, 17899) 0.5% Blocking reagent (Roche 11096176001) and 10 mM Tris, pH 7.4, for 3 min at 80 °C and then 2 h at RT, followed by 2 × 15 min washes in formamide 70%, 10 mM Tris, pH 7.4 and three washes with PBS-0.05% Tween 20. The coverslips were mounted using ProLong Gold (Invitrogen, P36930). Images were acquired in DV Elite system (GE Healthcare) equipped with a IX71 microscope (Olympus) and a sCMOS camera and driven by softWoRx version 7.0.0 or in a CSU spinning disk confocal microscope (Olympus) Camera Ixon Ultra 897 (Andor) Laser lines 405 nm, 488 nm, 561 nm, 640 nm Software cellSens Dimension 1.18.

All digital images were analyzed in FIJI/ImageJ software v2.0.0-rc-69/1.52p.

**Electron microscopy analysis**. EM analysis was performed according to[21]. Typically, 5 µl of telomere-enriched DNA corresponding to 5–20 ng were used for each spread. The DNA was mixed with 5 µl of formamide (Thermo Scientific, 17899) and 0.4 µl of benzalkonium chloride (BAC, Sigma B6285) 0.02% in TE. (BAC stock solution 0.2% in formamide was diluted 1:10 in TE 1X before use). After mixing, the drop was immediately spread on a water surface in a 15 cm dish containing 50 ml of distilled water, using a freshly-cleaved mica sheet (Ted pella inc, product no: 52-6) as a ramp. For non-enriched controls, 30 ng of KpnI-digested genomic DNA was spread using the same method. The monomolecular layer was gently touched with an EM grid prepared as follows. A thin (4–8 nm), homogeneous and low grain carbon layer was deposited on EM grids (Nanovision, PEG400). The carbon layer was created on a mica glass surface (2 cm × 2 cm) using the MED020 e-beam evaporator (Leica), equipped with the QSG monitor, two EK030 electron guns controlled by the EVM030 control unit. The e-beam evaporation parameters described in the instruction manual were used. The carbon layer deposited on the mica surface was floated on the surface of the water and transferred on the 400-mesh copper grids. Before use, carbon-coated EM grids were placed in contact with an Ethidium Bromide solution (33.3 µg/ml in H$_2$O) for 30–45 min at RT. Carbon grids with absorbed DNA molecules were immediately stained with a solution of uranyl acetate 0.2 µg/µl in ethanol and coated with 8 nm of Platinum using the MED020 evaporator modified with the low-angle grid shadowing kit (Leica 16770525) so that the sample holder was placed at an angle of 280.5 degrees and made an angle of around 3 degrees with the platinum gun fixed on the head of the instrument. For platinum e-beam evaporation, we utilized the parameters indicated in the MED020 instruction manual.

The DNA recovered from the linear and t-circle arc of 2D-gels, was spread using the droplet method as in[37]. Briefly, 1 ng of DNA in 28 µl of TE 1X, was mixed with 30 µl of Formamide and 2 µl of BAC 0.08% in TE1X. The droplet was incubated for 5 min at RT and the surface was gently touched with a carbon-coated EM grid, previously activated by contact with an ethidium bromide solution 33 µg/ml in TE 1X. The grids were then processed for staining with Uranyl Acetate and rotary shadowing as described in[21]. Kleinschmidt spreading was performed according to[20] with minor modifications. Briefly, 50 ng of DNA in TRIS 10 mM pH 8.0 was mixed with ammonium acetate (pH 7.8, 0.25 M final concentration). Cytochrome C (Sigma) was added to 4 µg/ml final concentration and the droplet (50 µl) was placed on parafilm for 90 s. A carbon-coated EM grid was touched to the drop and then dehydrated through two washes of 30 s in 75% and 90% ethanol, followed by air drying and rotary shadowing with platinum. TEM pictures were taken using a FEI Tecnai12 Bio twin microscope operated at 120 KV and equipped with a side-mounted GATAN Orius SC-1000 camera controlled by the Digital Micrograph software. For acquisition of large areas, overlapping fields were acquired and stitched using the Digital Micrograph software. Images in DM3 format were analyzed in FIJI/ImageJ software v2.0.0-rc-69/1.52p. In these conditions, in BAC spreads, a length of 0.36 µm corresponds to 1 kb of double-stranded DNA. Data annotation and storage during the analysis was performed with an Image J macro (Supplementary Software 1).

**DNase I treatment on isolated nuclei**. MEF nuclei were isolated according to[38]. Briefly, cells were collected by trypsinization, washed with ice-cold PBS, and resuspended in ice-cold fibroblast lysis buffer (12.5 mM Tris pH 7.4, 5 mM KCl, 0.1 mM spermine, 0.25 mM spermidine, 175 mM sucrose, supplemented with protease inhibitor cocktail (Roche, 11836170001) at a concentration of $8 × 10^6$ cells/ml). After 10 min of incubation on ice, 0.02 volumes 10% NP-40 was added and cells were incubated for 5 min on ice. Nuclei were collected by centrifugation at 1000 g for 5 min at 4 °C and washed once with ice-cold Nuclei Wash Buffer (NWB) (10 mM Tris-HCl pH 7.4, 15 mM NaCl, 60 mM KCl, 5 mM MgCl$_2$, 300 mM sucrose) and resuspended in NWB. When indicated, psoralen crosslinking was performed on the nuclei suspension in NWB, as described above for cell suspensions. For the DNase I treatment, 1 volume of nuclei suspension was mixed with 1 volume of DNase I cocktail (NWB supplemented with CaCl$_2$ 2 mM, BSA 100 µg/ml, and twice the indicated concentration of DNase I (Roche 10104159001) and incubated for 8 minutes at RT. The reactions were stopped with 0.5 volumes of ice-cold stop buffer (50 mM EDTA, 10 mM EGTA). The nuclei were then processed for genomic DNA extraction as described above for cells.

**DNase I treatment on isolated DNA**. Genomic DNA, extracted as described above, was incubated with DNase I (Roche 10104159001) in 10 mM Tris-HCl pH 7.4, 15 mM NaCl, 60 mM KCl, 5 mM MgCl$_2$, 10 mM CaCl$_2$, for 8 min at RT. The reaction was stopped by adding 0.2 volumes of EDTA-EGTA 0.25 M each, extracted with 1 volume of phenol-cholorform isoamylalcohol and precipitated in isopropanol.

**Two-dimensional agarose gel electrophoresis**. 10 µg of genomic DNA was digested overnight with 20 units of AluI and MboI (NEB) and then precipitated with isopropanol. For the analysis of the mouse L1 repeats, the DNA was digested either with BglI or with KpnI as indicated. The first dimension was run in 0.35% agarose (US-biological, A1015) in TBE 0.5X, without ethidium bromide for 12–24 h at 1 V/cm. The gel was stained with 0.3 µg/ml ethidium bromide in TBE 0.5X and lanes were excised above 5 kb for mouse, U2OS and HeLa 1.3 telomeres and above 2 kb for HeLa 204 and HTC75 telomeres. The second dimension was run in 0.7% agarose in TBE 0.5X with 0.3 µg/ml ethidium bromide at 3–5 Volts/cm at 4 °C. When necessary, psoralen crosslinking was reversed before Southern blotting by exposing the gel to 254 nm UV for 10 min in a Stratalinker (UVP CL1000 Ultra-violet crosslinker). For Southern blotting, the gel was first incubated 2 × 30 min with the depurination solution (HCl 0.25 N), 2 × 30 min with denaturing solution (NaOH 0.5 M, NaCl 1.5 M), 2 × 30 min with neutralizing solution (Tris 0.5 M pH 7.5, NaCl 3 M). The DNA was then transferred by capillarity in SSC 20X onto an Amersham Hybond-X membrane (GE healthcare RPN203). For TTAGGG repeats probe the 800 bp EcoRI fragment of the Sty11 plasmid (a gift from Titia de Lange)[39] was used. For the L1 repeats probe a 1 kb EcoRV fragment, containing mouse BamHI dispersed repeats[40] cloned in pBlue was used. Radioactive signal was captured on phosphor screens (FUJIFILM Storage Phosphor screen MS3543 E), read on a Typhon Trio (GE) and analyzed on ImageJ.

**Telomere blots**. 10 µg of genomic DNA was digested overnight with 20 units of AluI and MboI (NEB) and then precipitated with isopropanol. The DNA was separated in a 0.7% agarose gel (US-biological, A1015) in TBE 0.5X, with 0.3 µg/ml ethidium bromide for 21 h at 1.6 V/cm. The gel was then processed for Southern blotting and hybridization with a telomeric probe as described above.

**Incubation with HeLa extracts**. 1 µg of genomic DNA was incubated with 60 µg of HeLa nuclear extract (6 mg/ml) (IPRACELL, CC012010) in 50 mM Tris HCl pH8,

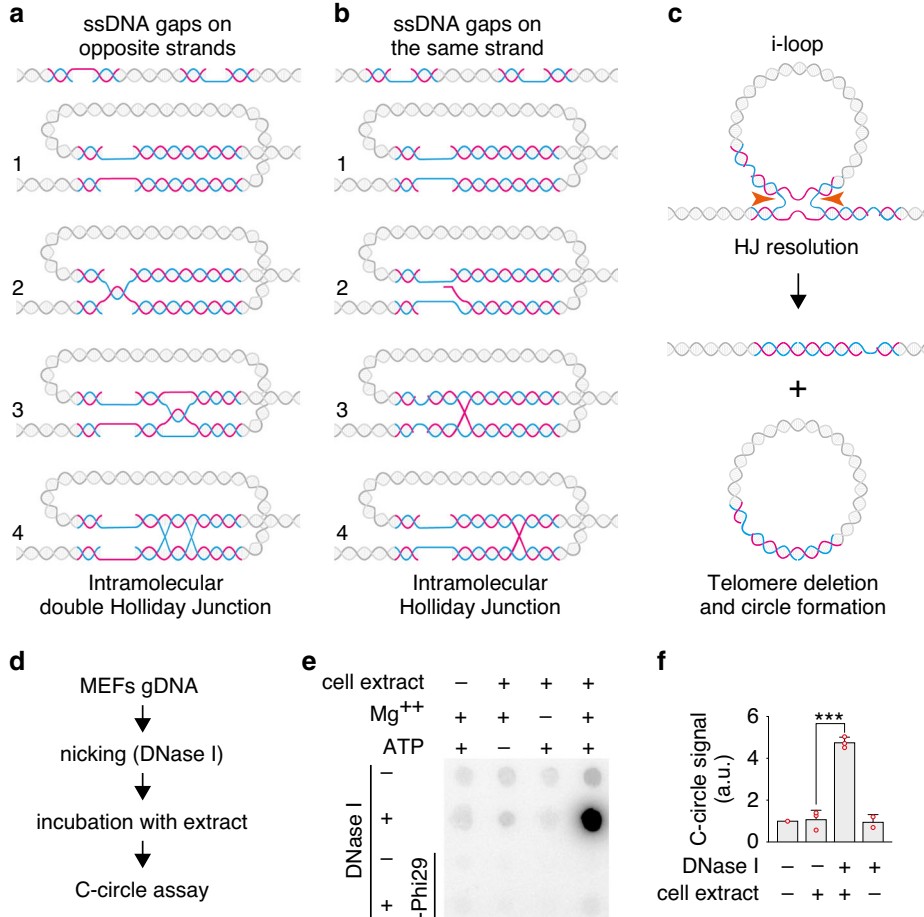

**Fig. 7 I-loops are a substrate for the generation of telomeric circles. a** Formation of i-loops, by gaps on opposite telomeric strands. Exposed complementary DNA can come in close proximity by looping and undergo base pairing (step 1). Plectonemic pairing can occur by strand rotation, resulting in an i-loop (step 2). The loop junction could branch migrate as a hemicatenane (step 3) that could be transformed in a double Holliday junction by pairing of the opposite strands (step 4). **b** Formation of i-loops, by gaps on the same telomeric strand. The gaps can come in close proximity by DNA looping (step 1) and promote an exchange of the complementary strands (step 2) resulting in an i-loop with a single Holliday junction at the base (step 3), that can undergo branch migration (step 4). **c** Generation of telomeric circles via i-loop excision by Holliday junction resolvases. Cleavage on the horizontal axis of the image (orange arrows) will result in the excision of the loop as a circle and telomere loss. Note that the excised circle, would contain a nick, resulting from the HJ resolution and one of the original single-stranded gaps that induced formation of the i-loop. **d** Experimental procedure to test the model shown in (**c**). Isolated SV40-LT MEFs genomic DNA was nicked with low concentrations of DNase I, which induces the formation of i-loops at telomeres. The DNA was extracted, incubated for 30 min at 37 °C with a HeLa nuclear extract to allow HJ resolution and the presence of telomeric circles was assayed with the C-circle assay. **e** Dot blot of the C-circle assay in (d) hybridized with a probe recognizing the telomeric repeats. Source data are provided as a Source Data File. **f** Quantification of the telomeric signal in dot blots from 3 independent experiments as the one described in **d**. Bars represent the mean with the standard deviation. Single data points are also shown as red dots. The signal is reported relative to the untreated sample (no DNase I, no extract) which was set to 1. *P* value = 0.0003, was derived from unpaired, two-tailed, Student's *t*-test. Source data are provided as a Source Data File.

150 mM NaCl, 5 mM MgCl2, 2 mM ATP, 1 mM DTT for 35 min at 37 °C in 20 μl final volume. The reaction was stopped with 0.1 volumes of EDTA-EGTA 0.25 M each, extracted with 1 volume of phenol-cholorform isoamylalcohol and precipitated in isopropanol.

**C-circle assay**. Was performed according to[6]. Briefly, 25 ng of genomic DNA, digested with AluI and MboI, were incubated for 12 h at 30 °C with 7.5 Units of Phi29 polymerase (NEB M0269) in Phi29 NEB buffer 1X, supplemented with dNTPs 0.37 mM each, in a final volume of 20 μl. The enzyme was inactivated by heating to 65 °C for 20 minutes and the reaction was blotted onto a Hybond-X membrane. Telomeric repeats were detected using the TTAGGG repeats probe described above.

**Statistics and reproducibility**. The telomere enrichment procedure (Fig. 1a–d, Supplementary Fig. 1a, b) was repeated more than 3 times for MEFs and more than 3 times for human cells (U2OS and HeLa) with similar results. EM analysis of i-loop and t-loop accumulation in telomere-enriched fractions of mouse telomeres (Fig. 2a–d, Supplementary Fig. 2a–c) was performed on 3 independent

experiments. High magnification analysis of i-loop junction (Fig. 3) was performed on a pool of 109 i-loops from three independent experiments. All 2D-gel experiments with and without DNase I treatment (Figs. 5a, b and 6a, b, Supplementary Fig. 6c, d) were performed at least 3 times, with similar results. The C-circle assay after incubation with the HeLa extract (Fig. 7e, f) was performed more than 3 times with similar results. EM analysis of ALT telomere structure (Fig. 4a–d, Supplementary Fig. 5a, b) was performed on more than 447 molecules/condition from one telomere enrichment and signal purification from 2D-gels. EM analysis of telomere structure after DNase I treatment (Fig. 5c–e) was performed on more than 206 molecules/condition from one DNase I treatment and telomere enrichment procedure. EM analysis of i-loop accumulation in HeLa 1.3 cells (Supplementary Fig. 3a–c) was performed on more than 239 molecules/condition from one telomere enrichment procedure. Comparison of i-loop frequency in Kleinschmidt vs BAC spreading (Supplementary Fig. 4a–d) was performed on more than 194 molecules/condition from one telomere enrichment procedure. DNase I treatment and 2D-gels in human cells (Supplementary Fig. 6b) were repeated twice with similar results. The telomere blot in Supplementary Fig. 6a was performed once as a control. Statistical analysis were performed with Prism (v6.0c, Graph Pad).

**Reporting summary**. Further information on research design is available in the Nature Research Reporting Summary linked to this article.

## Data availability
All data generated or analyzed during this study are included in this published article. Source data are provided with this paper. All raw EM images are available from the corresponding author upon reasonable request. Source data are provided with this paper.

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

## Acknowledgements
We are grateful to Titia de Lange for allowing Y.D. to initiate this project in her lab, for reagents, and for helpful discussions. We are grateful to Smaranda Willcox for her invaluable help with the Kleinschmidt spreading. We thank Marco Foiani and Fabrizio d'Adda di Fagagna for technical support and helpful discussions and Elia Zanella for helpful discussions. We are grateful to members of the IFOM Cell Biology Unit for their invaluable assistance with growing large cell cultures. We thank Paolo Maiuri for developing an ImageJ macro for the annotation and storage of the EM pictures and the IFOM imaging facility for technical assistance. YD lab is supported by the Associazione Italiana per la Ricerca sul Cancro, AIRC, IG 19901. F.P. is supported by the Fondazione Telethon GGP17111.

## Author contributions
A.H. and D.P. assisted with the setting of the sucrose gradient conditions for the telomere enrichment procedure. A.H. performed the U2OS telomere purification from 2D-gels. M.Giannattasio. provided technical assistance with the initial DNA spreading and EM procedure. M.Galli performed the EM experiment in HeLa 1.3 cells. F.P. assisted with the HeLa extract incubation. G.M. and Y.D. performed the rest of the experiments. Y.D. conceived the study and wrote the paper.

## Competing interests
The authors declare no competing interests.
