## [Peer Review File · Nature Communications]

Reviewers' comments:

Reviewer #1 (Remarks to the Author):

This paper from Doksani and colleagues has some very strong points including a new more rapid method for purifying telomeric DNA, and a different EM approach for imaging DNA, t-circles and telomeric DNA in general. Their ability to isolate large DNAs from the gels and image them by EM opens up many possibilities. The work argues that in ALT cells, the double strand circles that have been observed before by numerous laboratories actually result from internal loops that are cleaved out (i-circles). This hypothesis while interesting runs counter to some of the established work from several groups and further is based on the use of an EM preparative method that has the potential for generating internal loops in DNA. Thus as the work stands, it is in need of verification by other methods before the conclusions presented would reach a level of acceptance in the field.

General comments

1. The T-circle story has become confused and needs to be referenced and cited correctly. The original discovery showing free double stranded circles consisting of telomeric repeats in ALT cells was the paper of Cesare (2004) *Mol Cell Biol*, 24, 9948-9957. In that paper, it was shown that the size distribution of the free circles matched the size distribution of the circular portion of the t-loops in the same cells, leading to an obvious conclusion that the t-circles were the result of cleavage of the loop from the t-loop molecule. Boulton and colleagues have linked mutations in proteins such as RTEL to the release of the t-circles from the end of the telomere. Thus there are strong arguments for the model that the circular portion of the t-loop can be released in the form of t-circles. If internal loops are also cut out, this would be interesting, but would seem to be in addition to the release of terminal circles. In this report, the implication seems to be that all the t-circles come from internal loops and that seems to contradict much previous work and thinking.
2. The description of t-circles has recently become confused due to the Reddel assay which they called a C circle assay. T-circles come in a variety of forms including ones that are covalently closed on both strands, and ones that contain a nick or gap in either the C-rich strand or the G-rich strand. Reddel developed a rolling circle assay for t-circles based on using phi 29 polymerase to initiate replication on t-circles. In this assay he picked the C strand as the template and the assay probes for molecules in which the C strand is circular and lacks a nick or gap. He could as easily have picked the G strand but for the former was chosen for technical reasons. This assay was termed the C circle assay. In either case the assay detects gapped or nicked T-circles. Use of the term "C-circles" is misleading as it erroneously suggests that ALT cells are accumulating excess single stranded circles formed from telomeric repeats which is not the case. Indeed there is no such entity as a C-circle in a cell, only nicked, gapped or closed t-circles. Hence it is important to be clear on this point. The proper term should be either t-circles or gapped/nicked T-circles.
3. The real issue here is how to distinguish between: A) the interesting situation which these authors have raised in which the telomeric DNA is looped out with some form of strand exchange or strand pairing at the junction which then results in a cleavage event followed by the discharge of a telomeric circle and: B) the always present cross-overs of long double strand DNA as it is laid out over the EM grid. DNA is a long flexible molecule and while the surface forces at an air-liquid interface and the binding of the BAC or denatured protein tend to stiffen DNA, nonetheless long linear DNAs always show internal cross-overs. It is also possible that due to the unusual nature and sequence of telomeric DNA that this will occur more often with telomeric dsDNA than random sequence DNA. This concern is compounded by the spreading method used here (BAC/formamide) in which the high formamide concentration is more likely to result in some local breathing which could generate transient pairings as the DNA folds over itself following purification.
It seems difficult to distinguish between these two possibilities. The authors have made attempts but these efforts seem unlikely to convince anyone with an understanding of the normal artifacts of the preparation of DNA for EM, or DNA topology in general. There however two sets of experiments which would go a long way to support the conclusions of this study.
The first would be to prepare the DNAs using the very classic EM preparative method of Kleinschmidt in

which the DNA is spread on an air-water interface in a film of denatured cytochrome C protein. No formamide is included and DNA is spread from a solution of ammonium acetate. This method had been used by countless investigators for nearly 70 years and if it shows the same internal loops in the ALT cells this would help bolster the work presented here.

The second approach which Dr. Doksani is an expert in, would be to image these molecules directly by STORM light microscopy. There are other such light microscopic methods that are easier and have been used to image t-loops, so several options seem available. As Doksani himself was the one who elevated the t-loop field by using STORM, it seems obvious that they would complement their work by showing internal loops (if they exist) by this methods.

Minor point. Have the authors examined a series of different ALT lines and have they verified the nature of these lines by footprinting? ALT lines grow slowly and can be taken over by HeLa and other fast growing, non-ALT lines.

Reviewer #2 (Remarks to the Author):

In this manuscript, Mazzucco et al. describe the development of a two-step procedure for the purification of telomeric DNA from mammalian cells and the visualization of the purified fragments by Electron Microscopy (EM). Using this approach, the authors found that telomeric DNA fragments are enriched in internal loops (i-loops). These structures are particularly frequent in telomeres from cells that use the ALT mechanism to maintain telomeres. The authors also produce data indicating that i-loops form in correspondence of DNA damage and exposure of single-stranded DNA at telomeres. Further, they suggest that i-loops might represent an intermediate of extrachromosomal telomeric circle in ALT cells. The manuscript is of great interest for the scientific community. Experiments are rigorous and contribute important information to understand which structures form at telomeres in response to DNA damage and on telomere metabolism in ALT cells. The manuscript is overall clearly written and I recommend its publication on Nature Communications.

Minor remarks:

Figure 4 and Figure S4 show EM imaging of telomeric DNA molecules from ALT cells. Differently from the enrichment procedure described in M&M (page 19), telomeric DNA was purified from a 2D-gel, isolating separately molecules from the linear and the T-circle areas. The authors show a significant enrichment of circular molecules in the material isolated from the t-circle area (Fig. 4B, S4B). However, 39.4 % of DNA molecules from the t-circle area are linear. Could this reflect the disruption of part of the circular structures during purification? The second purification step of telomeric DNA from U2OS cells differs from that shown in Fig. 1B. DNA was stained with EtBr after the 1st dimension and separated in the presence of EtBr during the 2nd dimension. This procedure (followed by UV) might damage DNA. Did the authors perform EM imaging on U2OS telomeres purified as in Fig. 1 B and in this case which is the percentage of i-loops?

Figure 1A-B and Figure 4A. Possibly, lanes should be numbered.

Figure 2B shows that i-loops are present also in 3% of control genomic DNA. Did the authors find single-stranded regions at the junctions also in control as in Fig. 3 and at which frequency?

Reviewer #3 (Remarks to the Author):

This is a very interesting paper that has the potential to make a major contribution to understanding the origins of extrachromosomal telomeric DNA, which have been a conundrum for more than 20 years. The implications for other repetitive DNA are also very important, and should make this paper of great interest to a wide audience. The data are mostly convincing, and the paper is exceptionally well-written.

1. The EM data appears to be the strongest and most direct evidence for the existence of internal loops (i-loops) in telomeric DNA. (a) As a non-EM expert, this reviewer would like to be convinced that the images do not simply represent DNA molecules folded over themselves on the grid (rather than true looping-out events), which for some reason are more common in telomeric DNA. Could the authors please address this point. (b) In addition, is it possible that the frequent appearance of single-stranded DNA is a shading artefact of the EM staining produced at the point where the DNA is folded itself?
2. The DNA that is mostly used in this study is from SV40LT-immortalized MEFs. Do these cells have an activated telomere length maintenance mechanism and, if yes, do they utilize ALT or telomerase?
3. Methods, Cell culture. The sources of all cell lines need to be specified. The culture conditions for U2OS, HeLa204 and HTC75 cells need to be defined. It is essential to indicate when the cells were tested for Mycoplasma species and when cell authentication was performed.
4. Fig 1D: Could the authors comment on the likely reasons the telomeric DNA is so "blobby" in appearance? Is this a consistent feature of the enriched telomeric DNA preps when combed?
5. Figure S1B: would it be possible to provide higher magnification for all the panels in this figure? (Not essential, but may help to make the data a little more convincing.)
6. Fig. 7D – the figure legend needs to indicate the source of the DNA.
7. The experiments with DNase I may yield a very important insight into the mechanism of i-loop formation, but there are some aspects of the data that need to be addressed. (a) in Fig. 7E, it is essential to show the Phi29-negative controls. Although linear ss DNA is not amplified by Phi29, it does get copied and therefore linear ss telomeric DNA contributes to the signal in the C-circle assay. (b) Some aspects of Figure S5 seem unconvincing. In Panel A, it is not clear that there is an arc of circular DNA on the 2D gels that are shown for HeLa 1.3. Instead there are curves that are more like concentric arcs rather than the divergent arc that is typical of circular DNA. Is there an explanation for this? The other two cell lines in Panel A have typical circular DNA arcs, and the comparison between the untreated DNA and the DNA treated with the lowest amount of DNase I is convincing. The circular arcs do not appear to increase with increasing DNase I digestion, but they seem to decrease less than the arc of linear DNA does.
8. Has DNA from human ALT cells been treated with DNase I?

Minor

9. On p4 line 21, it needs to be specified that the DNA is from a mouse cell line ("SV40 MEFs), otherwise the subsequent comment about mouse BamH1 repeats is not readily understandable.
10. "BamH1 repeats" is a terminology that does not appear to have been used much since the 1980s.
11. References: The titles are a mix of sentence case and all words capitalised.
12. Ref. 15 – citation is incomplete (article number is missing).

We are grateful to the reviewers for the time dedicated to our manuscript and for their helpful suggestions and comments. Below, are our answers point by point.

Reviewer 1: This paper from Doksan and colleagues has some very strong points including a new more rapid method for purifying telomeric DNA, and a different EM approach for imaging DNA, t-circles and telomeric DNA in general. Their ability to isolate large DNAs from the gels and image them by EM opens up many possibilities. The work argues that in ALT cells, the double strand circles that have been observed before by numerous laboratories actually result from internal loops that are cleaved out (i-circles). This hypothesis while interesting runs counter to some of the established work from several groups and further is based on the use of an EM preparative method that has the potential for generating internal loops in DNA. Thus as the work stands, it is in need of verification by other methods before the conclusions presented would reach a level of acceptance in the field.

Authors: *The model proposed here is consistent with all previous work on telomeric circles and does not exclude the existence of the t-loop excision model. We believe that in many instances, excision of damage-induced i-loops proposed here, provides a better explanation for the accumulation of telomeric circles than the t-loop excision model. The latter was proposed in the context of a mutation in TRF2, the main factor involved in t-loop formation/stabilization; it assumes that the t-loop junction becomes deprotected in the TRF2 delta-basic mutant (Wang et. al., 2004 PMID: 15507207). Invoking the t-loop excision model implies a defect in t-loop protection. This should advice against its unconditioned use to explain accumulation of telomeric circles in tens of different mutants/settings that have nothing to do with t-loop metabolism. In order to be more explicit in this point, we modified the sentence in the introduction where we now refer to additional mechanisms of t-circle formation, instead of alternative mechanisms of t-circle formation. In the introduction section, page 3 line 17: **“These results suggest the existence of additional mechanisms of t-circle formation.”** In the discussion section, page 9, line 26: **“Our data provide an additional mechanism of t-circle formation, as a consequence of telomeric damage.”***

As for the potential of our EM preparative method in generating internal loops in DNA. We verified this experimentally by spreading the same DNA in parallel with the BAC-formamide method (used in our study) and the classic Kleinschmidt method. We did

observe a clear enrichment of i-loops at telomeres also with the Kleinschmidt method. In addition, the BAC-formamide spreads showed a substantially lower background of internal loops than Kleinschmidt spreads of the same DNA prep. These data are included in the Figure S4 of the revised manuscript and are further discussed below.

Reviewer 1: The T-circle story has become confused and needs to be referenced and cited correctly. The original discovery showing free double stranded circles consisting of telomeric repeats in ALT cells was the paper of Cesare (2004) Mol Cell, Biol, 24, 9948-9957. In that paper, it was shown that the size distribution of the free circles matched the size distribution of the circular portion of the t-loops in the same cells, leading to an obvious conclusion that the t-circles were the result of cleavage of the loop from the t-loop molecule.

Authors: *We agree with the reviewer that there is some confusion in the field. Indeed, Tony Cesare and Jack Griffith were the first to visualize by EM telomeric circles in ALT cells. Now we refer to this work also at the beginning of the introduction, when we first mention ALT. One point of confusion is the size of t-circles. The elegant EM work by Cesare and Griffith as well as our EM data, are both based on the purification of telomeric repeats by size exclusion (i.e. digesting genomic DNA and discarding most of the fragments smaller than 10 kb). This step obviously counterselects smaller circles, therefore, the size distribution of t-circles in ALT cells, obtained in these EM studies, may not represent the real size distribution of t-circles in ALT cells.*

Reviewer 1: Boulton and colleagues have linked mutations in proteins such as RTEL to the release of the t-circles from the end of the telomere. Thus there are strong arguments for the model that the circular portion of the t-loop can be released in the form of t-circles. If internal loops are also cut out, this would be interesting, but would seem to be in addition to the release of terminal circles. In this report, the implication seems to be that all the t-circles come from internal loops and that seems to contradict much previous work and thinking.

Authors: *It is not our intention to imply that all t-circles come from excision of i-loops. Rather, we propose i-loop excision as an additional model to explain accumulation of t-circles in conditions of telomere damage and not in conditions of impaired t-loop metabolism. Boulton's lab has shown that RTEL1 is recruited at telomeres by TRF2, likely to unwind t-loops during S-phase; it follows that RTEL1-deleted cells will have an impaired*

t-loop metabolism (Sarek et al., Nature 2019 PMID: 31723267; note that RTEL1 has also a TRF2-independent role in telomere replication). Although technically one cannot tell whether a circle was released from the end or from the middle of the telomere, the data that link RTEL1 with t-loop metabolism support a t-loop excision model in this specific case. In ALT cells however, TRF2 or RTEL1 functions are not impaired (Lovejoy et al 2012 PMID: 22829774), and there are no reports of a defective t-loop metabolism. Rather, there is accumulation of telomere damage and t-circles. In this case our model provides a more straightforward explanation of the data, compared to the t-loop excision model. Another example: while it is not clear why the t-loop would get excised if telomeres are over elongated by telomerase, which leads to formation of telomeric circles in HeLa cells (Pickett et. al., 2009 PMID: 19214183), it is reasonable to assume that the probability of telomere damage and i-loop formation increases with telomere length. Also in this case, we believe our model could provide a more cogent explanation of the results than the t-loop excision model. The same reasoning could be applied to the many conditions associated with accumulation of telomeric circles (e.g. see references 3,9-16 in the paper). In summary, the model that better fits the experimental data should be adopted in each case. To be more explicit in this point, we have now changed the text referring to an “additional mechanism of t-circle formation” both in the introduction (page 3, line 17) and in the discussion (page 9, line 26).

Reviewer 1: The description of t-circles has recently become confused due to the Reddel assay which they called a C circle assay. T-circles come in a variety of forms including ones that are covalently closed on both strands, and ones that contain a nick or gap in either the C-rich strand or the G-rich strand. Reddel developed a rolling circle assay for t-circles based on using phi 29 polymerase to initiate replication on t-circles. In this assay he picked the C strand as the template and the assay probes for molecules in which the C strand is circular and lacks a nick or gap. He could as easily picked the G strand but for the former was chosen for technical reasons. This assay was termed the C circle assay. In either case the assay detects gapped or nicked T-circles. Use of the term “C-circles” is misleading as it erroneously suggests that ALT cells are accumulating excess single stranded circles formed from telomeric repeats which is not the case. Indeed there is no such entity as a C-circle in a cell, only nicked, gapped or closed t-circles. Hence it is important to be clear on this point. the proper term should be either t-circles or gapped/nicked T-circles.

Authors: *We agree with the reviewer that the existence of many terms to referring to extrachromosomal telomeric circles is confusing. Indeed, we use “telomeric circles” or “t-circles” throughout the text, except when we refer to the accumulation of C-circles in ALT cells or to the use of the C-circle assay. To our knowledge, the wide use of the term C-circles in the field is based on the report by Reddel’s lab that C-circles are 100-fold more abundant than G-circles in ALT cells (Henson et al., 2009 PMID: 19935656). In the same work, it is shown that rolling circle amplification of telomeric circles in ALT cells is not significantly impaired if dCTP is not added to the reaction, suggesting that is mostly the C-strand that serves as a template for the amplification. The reason why ALT cells appear to have a lot more C-circles than G-circles is not clear.*

Reviewer 1: The real issue here is how to distinguish between: A) the interesting situation which these authors have raised in which the telomeric DNA is looped out with some form of strand exchange or strand pairing at the junction which then results in a cleavage event followed by the discharge of a telomeric circle and: B) the always present cross-overs of long double strand DNA as it is laid out over the EM grid. DNA is a long flexible molecule and while the surface forces at an air-liquid interface and the binding of the BAC or denatured protein tend to stiffen DNA, nonetheless long linear DNAs always show internal cross-overs. It is also possible that due to the unusual nature and sequence of telomeric DNA that this will occur more often with telomeric dsDNA than random sequence DNA. This concern is compounded by the spreading method used here (BAC/formamide) in which the high formamide concentration is more likely to result in some local breathing which could generate transient pairings as the DNA folds over itself following purification.

It seems difficult to distinguish between these two possibilities. The authors have made attempts but these efforts seem unlikely to convince anyone with an understanding of the normal artifacts of the preparation of DNA for EM, or DNA topology in general. There however two sets of experiments which would go a long ways to support the conclusions of this study.

The first would be to prepare the DNAs using the very classic EM preparative method of Kleinschmidt in which the DNA is spread on an air-water interface in a film of denatured cytochrome C protein. No formamide is included and DNA is spread from a solution of ammonium acetate. This method had been used by countless investigators for nearly 70

years and if it shows the same internal loops in the ALT cells this would help bolster the work presented here.

The second approach which Dr. Doksani is an expert in, would be to image these molecules directly by STORM light microscopy. There are other such light microscopic methods that are easier and have been used to image t-loops, so several options seem available. As Doksani himself was the one who elevated the t-loop field by using STORM, it seems obvious that they would complement their work by showing internal loops (if they exist) by this methods.

Authors: *Following the reviewer's suggestion, we asked whether the preferential accumulation of i-loops at telomeres is also visible when the DNA is spread with the Kleinschmidt/Cytochrome C method. We enriched telomeric DNA and spread it in parallel with the Kleinschmidt or the BAC method. The Kleinschmidt spreading was performed according to the protocol used in Jack Griffith's lab (Griffith et al., 1999 PMID: 10338214) and had been previously set up in our lab with the help of Smaranda Willcox from Jack Griffith's lab. We detected a substantial accumulation of i-loops at telomere-enriched samples also in Kleinschmidt (CytC) spreads. These data are included in the Figure S4, in the revised version of the manuscript. Notably, although both methods revealed accumulation of i-loops at telomeric repeats, the BAC-formamide spread (the one used throughout this study) resulted in a substantially lower background of internal loops compared to the Kleinschmidt method (Figure S4). The following statement was included in the revised version of the manuscript, page 5 line 21: **"Accumulation of i-loops at telomere-enriched samples was observed also when the DNA was spread with the Kleinschmidt method²², although the background level of internal loops was higher in this setting (Figure S4A-D)"**. This difference could be due to the presence of formamide in the BAC spread, which limits the tendency of the DNA to self-interact via short/transient base pairings, as well as to the difference in the spread area (the surface of the 50 μ l drop in Kleinschmidt spreads vs nearly half the surface of a 15-cm dish in the BAC spread). Thus, the BAC spreading appears as the most appropriate choice in our study.*

Regarding the imaging of i-loops in super-resolution fluorescence microscopy we agree that this approach would require considerably less material and work than electron microscopy (e.g. 1 dish vs 50 dishes of cells; immediate spreading of a cell lysate vs 10 days long enrichment of telomeric DNA and so on). As an additional support of the validity of the reviewer's suggestion, we would like to point out that starting from figure 1 of our

published work on t-loops (Doksani et al., 2013 PMID: 24120135), there are several molecules with structures compatible with the presence of an i-loop and we have frequently observed these structures in STORM images of telomeres (see Figure 1 for reviewers only). Therefore, large-enough i-loops are certainly visible in super-resolution microscopy. However, we cannot establish the background level of i-loop formation in this setting, because it is not possible to visualize control (non-telomeric) DNA. One can label all telomeres with a single FISH probe, but we are not aware of an appropriate labeling for bulk DNA, or for any other long repetitive sequence, in super-resolution microscopy. Without this critical control, one could only rely on EM for validation, while using super-resolution fluorescence microscopy to compare i-loop frequency at telomeres in different mutants, similar to what it was done for t-loops. Even this application however, may be hampered by the fact that i-loops are considerably smaller than t-loops. While t-loop size follows telomere size (with no preference for strand invasion sites throughout the telomeric repeats), i-loops have a median size of 1.6 kb. Thus, due to resolution limits and poor FISH signal (that affects more smaller structures), large part of bona fide i-loops may not be identifiable in super-resolution fluorescence microscopy.

To address the reviewer's concerns, we have also performed another important control to support the EM experiments of this study. We subjected genomic DNA through a mock enrichment procedure, where the restriction enzymes were omitted. We found 5.1 % of molecules with i-loops in this sample (N= 927), about 3-fold less than in the telomere-enriched samples, demonstrating that the accumulation of i-loops at telomeric repeats is not attributable to our telomere enrichment procedure. The following statement was introduced in the revised version of the manuscript page 5 line 17: **"Accumulation of i-loops at telomeres could not be attributed to the enrichment procedure as mouse genomic DNA subjected to a mock enrichment procedure (where the restriction enzymes were omitted) showed 5.1% of i-loops (N=927 molecules)."**

In addition, the following considerations can further address the reviewer's main concern:

-our bulk DNA control and the mock enrichment control demonstrate that the abundance of i-loops at telomeres reported here, is not just a spontaneous property of the DNA in EM spreads or one induced by our enrichment procedure, but rather a telomere-specific phenomenon.

-our 2D-gel data shown in this paper strongly argue against a telomere-specific EM spreading artefact. Figures 5A and 5B show an intense t-circle signal (compatible with the

migration of telomeres containing i-loops) that is induced upon DNA damage and occurs in the absence of telomeric circles (see Figure 7D, E).

-EM analysis of a DNaseI-treated sample (that accumulates the t-circle signal in 2D-gels) also shows a 3-fold accumulation of i-loops at telomeres compared to the untreated control (Figure 5C-E). The samples were prepared and spread in parallel and therefore this difference cannot be explained by a (telomere-specific) spreading effect.

-the t-circle arc, purified from the U2OS cells, clearly shows an enrichment in i-loops compared to the linear arc (Figure 4). Also this, argues again the spreading alone generating i-loops. Furthermore, this experiment shows that the t-circle arc, that we detect in 2D-gels, is indeed made largely of i-loops.

In conclusion, although these structures happen to look like random crossing of a long DNA molecule and although it is possible that a small fraction of them are indeed an inevitable background of random crossings, the phenomena of i-loop enrichment at telomeric repeats, unveiled in this study, is supported by a combined 2D-gel and EM approach, with the necessary controls that separate the effect of the EM spreading from the biological effect of i-loop accumulation at telomeres. One important point of our manuscript is that we demonstrate that the famous t-circle arc in 2D-gels, used for decades to detect accumulation of telomeric circles, can be formed by damage-induced i-loops, in the absence of telomeric circles.

Reviewer 1: Minor point. Have the authors examined a series of different ALT lines and have they verified the nature of these lines by footprinting? ALT lines grow slowly and can be taken over by HeLa and other fast growing, non-ALT lines.

Authors: *Our cell lines are authenticated by footprinting and tested for mycoplasma, when they are first introduced in the institute and at every amplification of the stock, by the IFOM Cell Biology Unit. The technical details of the authentication of the U2OS cells and of the mycoplasma tests used are now indicated in the Materials and methods section (page 19, line 13).*

Reviewer 2: In this manuscript, Mazzucco et al. describe the development of a two-step procedure for the purification of telomeric DNA from mammalian cells and the visualization of the purified fragments by Electron Microscopy (EM). Using this approach, the authors found that telomeric DNA fragments are enriched in internal loops (i-loops). These structures are particularly frequent in telomeres from cells that use the ALT mechanism to maintain telomeres. The authors also produce data indicating that i-loops form in correspondence of DNA damage and exposure of single-stranded DNA at telomeres. Further, they suggest that i-loops might represent an intermediate of extrachromosomal telomeric circle in ALT cells. The manuscript is of great interest for the scientific community. Experiments are rigorous and contribute important information to understand which structures form at telomeres in response to DNA damage and on telomere metabolism in ALT cells. The manuscript is overall clearly written and I recommend its publication on Nature Communications.

Authors: *We thank the reviewer for appreciating our work.*

Reviewer 2: Minor remarks: Figure 4 and Figure S4 show EM imaging of telomeric DNA molecules from ALT cells. Differently from the enrichment procedure described in M&M (page 19), telomeric DNA was purified from a 2D-gel, isolating separately molecules from the linear and the T-circle areas. The authors show a significant enrichment of circular molecules in the material isolated from the t-circle area (Fig. 4B, S4B). However, 39.4 % of DNA molecules from the t-circle area are linear. Could this reflect the disruption of part of the circular structures during purification?

Authors: *Yes, it is possible that there is some resolution of structures during the isolation from 2D-gels which could account, at least in part, for the fraction of linear molecules observed in EM. However, it is not uncommon in EM imaging to obtain lots of linear “contaminants”, from areas of 2D gels where there should be only structures, especially when dealing with genomic DNA. For example, a similar approach, in a study of meiotic recombination intermediates in yeast, yielded 60-80% of linear molecules isolated from an area of the gel where structured DNA migrates (Cromie et al 2006; PMID 17174892). This could be due to imperfect separation of large amounts of genomic DNA in agarose gels, and in part to structure resolution during DNA isolation. We have mentioned these possibilities in the revised text (page 6, line 23).*

Reviewer 2: The second purification step of telomeric DNA from U2OS cells differs from that shown in Fig. 1B. DNA was stained with EtBr after the 1st dimension and

separated in the presence of EtBr during the 2nd dimension. This procedure (followed by UV) might damage DNA. Did the authors perform EM imaging on U2OS telomeres purified as in Fig. 1 B and in this case which is the percentage of i-loops?

Authors: *We agree with the reviewer, that eventual DNA damage that could occur during the second-dimension and excision from the gel may induce some i-loops. However, the material isolated in the same way, from the linear arc of the same gel, shows 3-fold less i-loops compared to the one isolated from the t-circle arc. Thus, UV damage alone cannot explain the accumulation of i-loops in the t-circle arc.*

In two independent experiments the telomere enrichment procedure from U2OS cells yielded ~9 and 16% of i-loops at telomere-enriched samples and 1.7 and 2.5% in the control bulk DNA, showing that i-loops are enriched also at U2OS telomeres. This value is higher than the one in HeLa cells with long telomeres, but similar to the one obtained in MEFs, which have significantly longer telomeres. Because i-loop accumulation is strongly influenced by telomere length (see Figure S6B), direct comparison of i-loop frequency between cell lines with different telomere length may not be informative on the intrinsic propensity of each cell line to accumulate i-loops.

Reviewer 2: Figure 1A-B and Figure 4A. Possibly, lanes should be numbered.

Authors: *We have numbered the sucrose gradient fractions in the indicated panels.*

Reviewer 2: Figure 2B shows that i-loops are present also in 3% of control genomic DNA. Did the authors find single-stranded regions at the junctions also in control as in Fig. 3 and at which frequency?

Authors: *We performed high magnification imaging of i-loops in a control genomic DNA spread and found that about 18% of the i-loops showed a single stranded DNA region at, or near the loop junction compared to 45% of the i-loops in the telomeric samples. The following statement was introduced in the revised version of the manuscript, page 6, line 3: “**In bulk DNA samples, around 18% of the loops showed a single stranded region at (or near) the loop junction (N = 109 loops)**”. It is possible that some of these loops are formed at other abundant tandem repeats in the genome.*

Reviewer 3: This is a very interesting paper that has the potential to make a major contribution to understanding the origins of extrachromosomal telomeric DNA, which have been a conundrum for more than 20 years. The implications for other repetitive DNA are also very important, and should make this paper of great interest to a wide audience. The data are mostly convincing, and the paper is exceptionally well-written.

Authors: *We thank the reviewer for her/his support of our study.*

Reviewer 3: Point 1. The EM data appears to be the strongest and most direct evidence for the existence of internal loops (i-loops) in telomeric DNA. (a) As a non-EM expert, this reviewer would like to be convinced that the images do not simply represent DNA molecules folded over themselves on the grid (rather than true looping-out events), which for some reason are more common in telomeric DNA. Could the authors please address this point.

Authors: *The combination of the 2D-gel data with the EM analysis rules out the possibility that i-loops are a simply a tendency of the telomeric DNA to fold on itself. Indeed, we show in Figure 5A and B that an intense t-circle signal (compatible with the migration of telomeres containing i-loops) is induced upon DNA damage and occurs in the absence of telomeric circles (see Figure 7D, E). This experiment demonstrates that DNA damage induces a telomeric structure which is abundant and visible in 2D-gels. Then, in Figure 5C-E, we show that the telomeric DNA, isolated from a DNaseI-treated sample (the condition that accumulates the signal in 2D-gels) has 3-times more i-loops than the untreated control, which was prepared and spread exactly in the same way. This argues against i-loops being formed during DNA spreading. Finally, we show that the “t-circle” arc purified from 2D-gels of U2OS telomeres is highly enriched in i-loops compared to the linear arc isolated and spread exactly in the same way (Figure 4). This result as well argues against i-loops being formed during the spreading and further demonstrates that the t-circle arc, that we detect in 2D-gels, is indeed made largely of i-loops. In connection to this point, we have also included in the revised version of the manuscript an additional control that further supports the EM experiments of this study. We subjected genomic DNA through a mock enrichment procedure, where the restriction enzymes were omitted. We found 5.1% of molecules with i-loops in this sample (N= 927), about 3-fold less than in the telomere-enriched samples, demonstrating that the accumulation of i-loops at telomeric repeats is not attributable to our telomere enrichment procedure. The following statement was introduced in the revised manuscript page 5, line 17: **“Accumulation of i-loops at telomeres could not be attributed to the enrichment procedure as mouse genomic***

DNA subjected to a mock enrichment procedure (where the restriction enzymes were omitted) showed 5.1% of i-loops (N=927 molecules).” Furthermore, in response to a request from reviewer 1, we also performed EM using an alternative DNA spreading procedure (Kleinschmidt/CytC) and show that a clear i-loop enrichment at telomeres is observed also in Kleinschmidt spreads, although the background level of internal loops is substantially higher in this setting. These data are included in Figure S4 of the revised manuscript.

Reviewer 3: (b) In addition, is it possible that the frequent appearance of single-stranded DNA is a shading artefact of the EM staining produced at the point where the DNA is folded itself?

Authors: Rotary shadowing is performed with the EM grid rotating at constant speed under the evaporating Pt rod, which ensures homogenous shadowing. If the appearance of the single-stranded DNA at the junction were due to uneven shadowing at the junctions we would expect thinner DNA regions at all junctions, which is not the case. Half of telomeric i-loop junctions and over 80% of bulk DNA i-loop junctions do not appear single-stranded.

Reviewer 3: Point 2. The DNA that is mostly used in this study is from SV40LT-immortalized MEFs. Do these cells have an activated telomere length maintenance mechanism and, if yes, do they utilize ALT or telomerase?

Authors: Mouse cells have long telomeres (5-10 times longer than human) that are maintained by telomerase. MEFs have an active telomerase pathway that is maintained after SV40 immortalization (Blasco et. al., 1997 PMID: 9335332; Margalef et al 2018 PMID: 29290468). SV40-MEFs do not show typical ALT hallmarks, like accumulation of APBs (ALT-associated PML bodies) or high levels of TSCEs (Telomere Sister Chromatid Exchange) (Lovejoy et al 2020 PMID: 31895940).

Reviewer 3: Point 3. Methods, Cell culture. The sources of all cell lines need to be specified. The culture conditions for U2OS, HeLa204 and HTC75 cells need to be defined. It is essential to indicate when the cells were tested for Mycoplasma species and when cell authentication was performed.

Authors: The sources for all the cell lines, along with the culture conditions are now specified in the Methods section. Our cell lines are authenticated by footprinting and tested for mycoplasma, when they are first introduced in the institute and at every amplification of the stock, by the IFOM Cell Biology Unit. The technical details of the authentication of the

U2OS cells and of the mycoplasma tests used are now indicated in the Materials and methods section (page 19, line 13). In addition, as a further control for HeLa1.3, HeLa204 and HTC75 cell lines, we included a new panel (A) in Figure S6 of the revised manuscript, with a blot showing the telomere length of the cell lines analyzed by 2D-gels. We want to point out that in the original Figure S5A, telomere lengths for the HeLa 204 and HTC75 cell lines (~2 kb and ~4kb) indicated the shortest telomeres in these cells, rather than the average telomere length. In the revised version of the figure (S6B in the revised manuscript) these values were corrected to 2-6 kb and 4-8 kb respectively, which indicate the telomere length range in these cells. This correction does not affect in any way the conclusions of the experiment.

Reviewer 3: Point 4. Fig 1D: Could the authors comment on the likely reasons the telomeric DNA is so "blobby" in appearance? Is this a consistent feature of the enriched telomeric DNA preps when combed?

Authors: *We attribute the blobby appearance of the telomeric DNA to imperfect stretching of some molecules on the glass coverslips, probably because the DNA is not prepared in the standard way for DNA combing (i.e. in agarose plugs). Because the yield of enriched telomeric DNA is very low and the material is very precious, we use smaller volumes for combing, compared to non-enriched genomic DNA. This could explain why blobs occur more frequently at telomere-enriched samples. However, this is not a very consistent feature. Even within the same preparation we have had a significant variability in the quality of the fiber stretching.*

Reviewer 3: Point 5. Figure S1B: would it be possible to provide higher magnification for all the panels in this figure? (Not essential, but may help to make the data a little more convincing.)

Authors: *We have now modified Figure S1B, showing a smaller area at a higher magnification.*

Reviewer 3: Point 6. Fig. 7D – the figure legend needs to indicate the source of the DNA.

Authors: *We have specified “isolated SV40-LT MEFs genomic DNA” in the revised figure legend. Panel 7D was also modified following the reviewer’s suggestion,*

indicating “MEFs gDNA” in the first line and “mouse genomic DNA” was inserted in page 9, line 13 of the revised version of the manuscript.

Reviewer 3: Point 7. The experiments with DNase I may yield a very important insight into the mechanism of i-loop formation, but there are some aspects of the data that need to be addressed. (a) in Fig. 7E, it is essential to show the Phi29-negative controls. Although linear ss DNA is not amplified by Phi29, it does get copied and therefore linear ss telomeric DNA contributes to the signal in the C-circle assay.

Authors: *We have modified the panel 7E by including a -Phi29 control, showing that the amplification that we see in the DNaseI-treated + extract incubation sample is not due to an excess of linear single-stranded DNA compared to the other samples.*

Reviewer 3: (b) Some aspects of Figure S5 seem unconvincing. In Panel A, it is not clear that there is an arc of circular DNA on the 2D gels that are shown for HeLa 1.3. Instead there are curves that are more like concentric arcs rather than the divergent arc that is typical of circular DNA. Is there an explanation for this?

Authors: *The explanation lies in the way that 2D-gel was run. In the particular gel shown in the original Figure S5A, the first dimension was run longer than usual and cut at higher molecular weights. This makes the second dimension appear less curved down at low MW. It could be seen as a technical “zoom in” at the high MW regions of other gels. Furthermore, this condition allows to separate further the structured DNA from the linear DNA in the second dimension. In order to avoid any misunderstanding, we now show in Figure S6B of the revised manuscript HeLa1.3 2D-gels that are less separated in the first dimension and show more downward pointing arcs. We would like to clarify that we do not think there is any circular DNA in those samples. This signal is induced by DNaseI treatment and our EM results show it’s made of i-loops.*

Reviewer 3: The other two cell lines in Panel A have typical circular DNA arcs, and the comparison between the untreated DNA and the DNA treated with the lowest amount of DNase I is convincing. The circular arcs do not appear to increase with increasing DNase I digestion, but they seem to decrease less than the arc of linear DNA does.

Authors: *Indeed, the main point of this experiment is that i-loop formation efficiency decreases significantly with telomere length. If telomeres are shorter the probability of having 2 or more sites of damage (that could interact and form an i-loop) is lower. For example, compared to HeLa1.3 one has to deliver 5-10 times more damage in order to*

have the same density of nicks at telomeres, and by doing so, you start inducing more and more DSBs and digesting away the DNA.

Reviewer 3: Point 8. Has DNA from human ALT cells been treated with DNase I?

Authors: *No.*

Minor

Reviewer 3: Point 9. On p4 line 21, it needs to be specified that the DNA is from a mouse cell line ("SV40 MEFs), otherwise the subsequent comment about mouse BamH1 repeats is not readily understandable.

Authors: *We specify "mouse genomic DNA" in the text*

Reviewer 3: Point 10. "BamH1 repeats" is a terminology that does not appear to have been used much since the 1980s.

Authors: *We now refer to mouse L1 repeats instead of BamHI repeats.*

Reviewer 3: Point 11. References: The titles are a mix of sentence case and all words capitalised.

Authors: *We have reformatted the references*

Reviewer 3: Point 12. Ref. 15 – citation is incomplete (article number is missing).

Authors: *We have reformatted the references*

[REDACTED]

See Figure 1D in Doksani et., al., *Cell*. 2013 Oct 10; 155(2): 345–356

REVIEWERS' COMMENTS:

Reviewer #1 (Remarks to the Author):

Review:

Telomere damage induces internal loops that generate telomeric circles
Mazzucco, Huda, Galli, Piccini, Giannattaso, Pessina and Doksani

Overview: this is a significantly improved work which now provides a compelling argument that DNA damage at mammalian telomeres which generate single strand gaps or breaks can lead to the formation of internal DNA loops. These loops appear to be generated by strand slippage and HR mechanisms that would occur in highly repetitive DNA. This study confirms the previous findings that double strand circles formed from telomeric repeats can be found in the "t-circle" arc in 2D gels, a finding first described years ago by Cesare et al in ALT cells and confirmed by Wang et al. The discovery by Doksani et al that the telomeric DNA in ALT cells frequently also contains internal loops which were linked to the presence of single strand gaps as seen by EM is a new important step in understanding how telomeres are maintained in ALT and the mechanisms involved. Why there would be this large accumulation of single strand gaps in ALT cells was not discussed. The results also seem to run counter to the current thinking and evidence that the t-circles in ALT result from cleavage of the circular portion of the t-loop, a model supported by data from a number of studies. Very straightforward and clever experiments using DNase1 were presented that bolster the in vivo findings. The quality of the EM images is excellent, in particular in the use of the BAC method. The new approach first described here to isolate telomeric DNA will be valuable in the field. Overall this work is worthy of publication in Nature and the issues raised here will stimulate discussion and experiments by numerous workers in the field.

Minor issues:

The English writing is quite well done however one more editing by an expert in English usage would be worthwhile ("in EM" should be changed to "by EM").

Page 3: lines 8-10. The discussion of t-circles is better in this revised ms, however it is still not quite right. The recent review of Tomaska et al has a nice short summary which might be worth looking at. In specific, it is now thought that the family of circles in ALT cells consists of double strand circles some of which would be covalently closed, some of which have nicks or gaps in either the C-rich or G-rich strand, and finally there appears to be a swarm of quite small fully single strand circles of the G-rich or C-rich strand, however the C-rich circles may be more prevalent. Some investigators have clouded the field by making it appear as if ALT cells specifically generate small C-rich circles which is not the case. Rather the rolling circle assay is designed to detect that species from among a family of different circles. Anything that these authors can do to make a clear statement about circle in ALT will aid the general readers.

Page 5; lines 15-25. Please cite the percentages of i-loops for the DNA spread by the Kleinschmidt method and also HeLa 1.3 with long telomeres.

Page 6: Line 4. It was not clear what was meant by bulk samples. Is this the genomic DNA?

Line 14. Cite reference 5 as well.

Line 23. Please expand on what DNA structures refers to.

Page 7: Line 5. It was not clear how the authors came to the conclusion that i-loops occur in proximity to DNA damage in ALT samples. This is a broad statement and needs to either be deleted or the rationale given.

Page 9: Line 15-16. For the general reader please explain the rolling circle assay. The reviewer assumes this is the phi-29 polymerase assay developed originally by a Czech group.

At some point in the paper a mention of Fouche et al NAR, 2006 where it was shown that replication forks in telomeric repeat arrays have a high propensity for fork slippage, and chicken foot generation might help bolster the model presented here.

Reviewer #2 (Remarks to the Author):

The authors have convincingly addressed all comments.

Reviewer #3 (Remarks to the Author):

The authors indicate in their rebuttal that all cell lines were authenticated upon receipt at IFOM, but the revised manuscript makes this statement regarding U2OS only.

Otherwise, I consider all comments to have been addressed satisfactorily by the authors.

Optional addition. The authors could consider the consequences of the increased content of non-canonical sequences in ALT telomeres. (a) The restriction enzymes used for the telomere enrichment procedure (e.g. Mnl1) may cut some of the non-canonical sequences. This does not affect the validity of the conclusions; if anything, it means that the number of i-loops detected in ALT cells is an under-representation. (b) The presence of non-canonical sequences may hinder branch migration once i-loops have formed, and could potentially contribute to the increased number of gapped t-circles in ALT cells that are detected by the C-circle assay. If the authors agree with this suggestion, they may wish to include it in the discussion of their model.

Roger Reddel

Point by point answers to reviewers' comments

REVIEWERS' COMMENTS:

Reviewer #1 (Remarks to the Author):

Review:

Telomere damage induces internal loops that generate telomeric circles
Mazzucco, Huda, Galli, Piccini, Giannattaso, Pessina and Doksani

Overview: this is a significantly improved work which now provides a compelling argument that DNA damage at mammalian telomeres which generate single strand gaps or breaks can lead to the formation of internal DNA loops. These loops appear to be generated by strand slippage and HR mechanisms that would occur in highly repetitive DNA. This study confirms the previous findings that double strand circles formed from telomeric repeats can be found in the “t-circle” arc in 2D gels, a finding first described years ago by Cesare et al in ALT cells and confirmed by Wang et al. The discovery by Doksani et al that the telomeric DNA in ALT cells frequently also contains internal loops which were linked to the presence of single strand gaps as seen by EM is a new important step in understanding how telomeres are maintained in ALT and the mechanisms involved. Why there would be this large accumulation of single strand gaps in ALT cells was not discussed. The results also seem to run

counter to the current thinking and evidence that the t-circles in ALT result from cleavage of the circular portion of the t-loop, a model supported by data from a number of studies. Very straightforward and clever experiments using DNase1 were presented that bolster the in vivo findings. The quality of the EM images is excellent, in particular in the use of the BAC method. The new approach first described here to isolate telomeric DNA will be valuable in the field. Overall this work is worthy of publication in Nature and the issues raised here will stimulate discussion and experiments by numerous workers in the field.

Authors: We thank the reviewer for the time dedicated to our manuscript and for the precious suggestions that helped improve our study.

Minor issues:

The English writing is quite well done however one more editing by an expert in English usage would be worthwhile (“in EM” should be changed to “by EM”).

Authors: The revised version of the manuscript includes the correction indicated by the reviewer, as well as other edits, kindly suggested by an expert in English usage.

Page 3: lines 8-10. The discussion of t-circles is better in this revised ms, however it is still not quite right. The recent review of Tomaska et al has a nice short summary which might be worth looking at. In specific, it is now thought that the family of circles in ALT cells consists of double strand circles some of which would be covalently closed, some of which have nicks or gaps in either the C-rich or G-rich strand, and finally there appears to be a swarm of quite small fully single strand circles of the G-rich or C-rich strand, however the C-rich circles may be more prevalent. Some investigators have clouded the field by making it appear as if ALT cells specifically generate small C-rich circles which is not the case. Rather the rolling circle assay is designed to detect that species from among a

family of different circles. Anything that these authors can do to make a clear statement about circle in ALT will aid the general readers.

Authors: We now refer to the very recent review from Tomaska et al., which, indeed, provides a beautiful historical perspective of the discovery of t-loops and telomeric circles. As for the distinction between C-circles and G-circles, we agree with the reviewer that there is some confusion in the field, but, since this is not a topic that we address in this paper, we feel that stating the topic and discussing it, might distract from the main message of the paper.

Page 5; lines 15-25. Please cite the percentages of i-loops for the DNA spread by the Kleinschmidt method and also HeLa 1.3 with long telomeres.

Authors: These values are now cited also in the main text.

Page 6: Line 4. It was not clear what was meant by bulk samples. Is this the genomic DNA?

Authors: Yes, it is genomic DNA. In the revised manuscript we specify bulk genomic DNA throughout the text

Line 14. Cite reference 5 as well.

Authors: Done.

Line 23. Please expand on what DNA structures refers to.

Authors: We now list in the text all the types of structures observed.

Page 7: Line 5. It was not clear how the authors came to the conclusion that i-loops occur in proximity to DNA damage in ALT samples. This is a broad statement and needs to either be deleted or the rationale given.

Authors: The sentence has been corrected and now reads as follows:
“Moreover, similar to mouse telomeres, close inspection of i-loops in ALT telomeres revealed that they often occurred in proximity of strand damage (red arrows in Figure 4d).”

Page 9: Line 15-16. For the general reader please explain the rolling circle assay. The reviewer assumes this is the phi-29 polymerase assay developed originally by a Czech group.

Authors: When we introduce the RCA assay for the detection of the circles in this paper, we refer to the Zellinger et. al., 2007 paper (K. Riha group), however, since our RC Assay was performed in the absence of a primer, as the one described in Henson et al 2009, we refer to this paper in that sentence. Now we state in the text that our RC Assay is done in the absence of a telomeric primer.

At some point in the paper a mention of Fouche et al NAR, 2006 where it was shown that replication forks in telomeric repeat arrays have a high propensity for fork slippage, and chicken foot generation might help bolster the model presented here.

Authors: We thank the reviewer for pointing this out to us. We now refer to this paper in the discussion, when we address the potential consequences of i-loop formation on telomere replication

Reviewer #2 (Remarks to the Author):

The authors have convincingly addressed all comments.

Reviewer #3 (Remarks to the Author):

The authors indicate in their rebuttal that all cell lines were authenticated upon receipt at IFOM, but the revised manuscript makes this statement regarding U2OS only.

Authors: We apologize for the confusion.

U2OS cell lines, that were provided and maintained by the IFOM cell biology unit, have been authenticated upon receipt and upon every amplification of the stock.

The two different HeLa clones; (HeLa1.3 and HeLa204) as well as the HTC75 cell lines were confirmed based on the most relevant criteria of this study: telomere length (Supplementary Figure 6A).

Our SV40-MEFs are clearly distinguishable from human cell lines used in this study and are anyway confirmed as mouse cell lines in Metaphase spreads and telomere length is verified in telomere blots.

All cell lines are tested for mycoplasma when they are first introduced in the institute and at every amplification of the stock by the IFOM cell biology unit.

Otherwise, I consider all comments to have been addressed satisfactorily by the authors.

Optional addition. The authors could consider the consequences of the increased content of non-canonical sequences in ALT telomeres. (a) The restriction enzymes used for the telomere enrichment procedure (e.g. Mnl1) may cut some of the non-canonical sequences. This does not affect the validity of the conclusions; if anything, it means that the number of i-loops detected in ALT cells is an under-representation. (b) The presence of non-canonical sequences may hinder branch migration once i-loops have formed, and could potentially contribute to the increased number of gapped t-circles in ALT cells that are detected by the C-circle assay. If the authors agree with this suggestion, they may wish to include it in the discussion of their model.

Roger Reddel

Authors: We thank Dr. Reddel for this suggestion. We agree that variant repeats could prevent extensive branch migration of i-loops and we now mention this scenario in the discussion, in the part that addresses i-loop resolution by branch migration. We also refer to the Conomos et. al., 2012 paper that reports the presence of variant repeats in ALT

cells.